# CONFORMAL PREDICTION WITH CORRUPTED LABELS: UNCERTAIN IMPUTATION AND ROBUST RE-WEIGHTING

**Shai Feldman**
Department of Computer Science
Technion, Israel
shai.feldman@cs.technion.ac.il

**Stephen Bates**
Department of Electrical Engineering
and Computer Science
Massachusetts Institute of Technology
stephenbates@mit.edu

**Yaniv Romano**
Departments of Electrical and Computer Engineering
and of Computer Science
Technion, Israel
yromano@technion.ac.il

## ABSTRACT

We introduce a framework for robust uncertainty quantification in situations where labeled training data are corrupted, through noisy or missing labels. We build on conformal prediction, a statistical tool for generating prediction sets that cover the test label with a pre-specified probability. The validity of conformal prediction, however, holds under the i.i.d assumption, which does not hold in our setting due to the corruptions in the data. To account for this distribution shift, the privileged conformal prediction (PCP) method proposed leveraging privileged information (PI)—additional features available only during training—to re-weight the data distribution, yielding valid prediction sets under the assumption that the weights are accurate. In this work, we analyze the robustness of PCP to inaccuracies in the weights. Our analysis indicates that PCP can still yield valid uncertainty estimates even when the weights are poorly estimated. Furthermore, we introduce *uncertain imputation* (UI), a new conformal method that does not rely on weight estimation. Instead, we impute corrupted labels in a way that preserves their uncertainty. Our approach is supported by theoretical guarantees and validated empirically on both synthetic and real benchmarks. Finally, we show that these techniques can be integrated into a triply robust framework, ensuring statistically valid predictions as long as at least one underlying method is valid.

## 1 INTRODUCTION

Modern machine learning models are increasingly deployed in high-stakes settings where reliable uncertainty quantification is essential. This need becomes even more critical when dealing with imperfect training data, which may be affected by noisy or missing labels. A common strategy to quantify prediction uncertainty is to construct prediction sets that cover the true outcome with a user-specified probability, e.g., 90%. Conformal prediction (CP) (Vovk et al., 2005) is a powerful framework for constructing theoretically valid prediction sets. Given a predictive model, CP utilizes labeled holdout calibration samples to compute prediction errors, which are then used to construct prediction sets for unseen test points. To provide a validity guarantee, this procedure requires that the training and test data are exchangeable, an assumption that does not hold in many real-world scenarios in which the observed data is corrupted.

To illustrate the challenge of providing reliable inference under corrupted data, we conduct an experiment on the *Medical Expenditure Panel Survey* (MEPS) (meps_19) dataset where the goal is to predict an individual's medical utilization ($Y \in \mathcal{Y}$) given a set of features ($X \in \mathcal{X}$), such as race, income, medical conditions, and other demographic variables. We simulate a missing-at-random setup by randomly removing labels with a probability that depends on the features. Since we cannot

use the samples with missing labels to compute the prediction error, we naively employ `CP` using only the observed data to construct prediction sets aiming to cover the true $Y$ with 90% probability. Figure 1 shows that this `Naive CP` fails to achieve the desired coverage due to the distributional shift induced by the missing labels.

However, in this special case, the distributional shift is a covariate shift which we can account for using the method of *weighted conformal prediction* (`WCP`) (Tibshirani et al., 2019). This method weights the data distribution by the labels' likelihood ratio so that the train and test points will look exchangeable. Figure 1 reveals that `WCP` applied with the true weights attains the desired $1 - \alpha = 90\%$ coverage rate, as theoretically guaranteed in (Tibshirani et al., 2019). Nevertheless, `WCP` requires all test features $X$ to be observed to compute the weights. This assumption might not hold in practice, e.g., when an individual does not share sensitive attributes at test time, such as their income or race in our MEPS example, due to privacy concerns. In such cases, `WCP` is infeasible and cannot be employed as it is impossible to compute the weights.

The goal of this work is to provide reliable inference under the setup of corrupted labels with missing features at test time (Collier et al., 2022; Wu et al., 2021; Ortiz-Jimenez et al., 2023). We refer to these features as *privileged information* (PI) (Vapnik & Vashist, 2009)—additional information available during training but unavailable at test time. In our MEPS example, the privileged features are race, income, an individual's rating of feeling, and more; see Appendix B for more practical examples of PI. *Privileged conformal prediction* (`PCP`) (Feldman & Romano, 2024) is a recent novel calibration scheme that builds on `WCP` to generate theoretically valid prediction sets without access to the test PI. However, `PCP` assumes access to the true weights of `WCP`, a requirement that might not hold in real-world applications. Indeed, Figure 1 shows that when applied with estimated weights, `PCP` does not achieve the nominal coverage level on the MEPS dataset. In this work, we focus precisely on this gap: what if the true weights are unavailable?

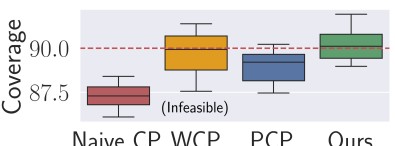

Figure 1: Coverage rate obtained on the MEPS19 data.

## 1.1 OUR CONTRIBUTION

This work provides two key contributions. First, we analyze the robustness of `PCP` and `WCP` to inaccuracies in the approximated weights. We formally characterize the conditions under which these methods maintain valid coverage despite the errors of the weights. In contrast with prior work (Lei & Candès, 2021; Bhattacharyya & Barber, 2024; Gui et al., 2024; Marmarelis et al., 2024) that deal with a worst-case analysis, our study reveals that `PCP` and `WCP` may construct prediction sets that attain the desired coverage rate even under significant errors in the weights, as demonstrated in empirical simulations in Section 3.1.2. Hence, the formulations developed in this work offer new theoretical guarantees and practical insights for these methods.

Second, we propose *uncertain imputation* (`UI`)—a novel calibration scheme that generates theoretically valid prediction sets in the presence of corrupted labels. In contrast to `WCP` and `PCP`, which assume that the weights can be estimated well from the PI, here, we assume that the clean labels can be estimated well from the PI (Wu et al., 2021; Xu et al., 2021; Collier et al., 2022). Under several assumptions, we show how to impute corrupted labels in a way that preserves the uncertainty of the imputed labels. By leveraging recent results on label-noise robustness of conformal prediction (Einbinder et al., 2023; Sesia et al., 2024), we theoretically show that our uncertainty-preserving imputation guarantees the validity of our proposal even when the weights are unreliable. Importantly, `PCP` might fail to achieve the nominal coverage level in such cases. Indeed, Figure 1 demonstrates that our proposed `UI` achieves the desired coverage rate while `PCP` does not.

Finally, we leverage the complementary validity conditions of `PCP`, `UI`, and `CP`, and propose combining all three into a triply robust calibration scheme (`TriplyRobust`) that constructs valid prediction sets when the assumptions of one of the methods are satisfied. Lastly, we conduct experiments on synthetic and real datasets to demonstrate the effectiveness of our proposed methods. Software implementing the proposed method and reproducing our experiments is available at `https://github.com/Shai128/ui`

## 1.2 PROBLEM SETUP

Suppose we are given $n$ training samples $\{(X_i, \tilde{Y}_i, Z_i, M_i)\}_{i=1}^n$, where $X_i \in \mathcal{X}$ denotes the observed covariates, $\tilde{Y}_i \in \mathcal{Y}$ the observed, potentially corrupted, labels, $Z_i \in \mathcal{Z}$ the PI, and $M_i \in \{0, 1\}$ is the corruption indicator. Specifically, if $M_i = 0$ then $\tilde{Y}_i = Y_i$, where $Y_i$, is the clean, ground truth label. Otherwise, if $M_i = 1$, then $\tilde{Y}_i$ is corrupted. For instance, in a missing response setup, if $M_i = 1$ then $\tilde{Y}_i = $ 'NA'. At test time, we are given the test features $X^{\text{test}} = X_{n+1}$, and our goal is to construct a prediction set $C(X^{\text{test}}) \subseteq \mathcal{Y}$ that covers the test response $Y^{\text{test}} = Y_{n+1}$ at a user-specified probability $1 - \alpha$, e.g., 90%:

$$\mathbb{P}(Y^{\text{test}} \in C(X^{\text{test}})) \geq 1 - \alpha. \tag{1}$$

This property is called *marginal coverage*, as the probability is taken over all samples $\{(X_i, \tilde{Y}_i, Y_i, Z_i, M_i)\}_{i=1}^{n+1}$, which are assumed to be drawn exchangeably (e.g., i.i.d.) from $P_{X, \tilde{Y}, Y, Z, M}$. The primary challenge in obtaining (1) is the distributional shift between the training data $\{(X_i, \tilde{Y}_i)\}_{i=1}^n$ and the test pair $(X_{n+1}, Y_{n+1})$. In practice, naively applying CP on the corrupted data could lead to unreliable uncertainty estimates (Barber et al., 2023), or overly conservative prediction sets (Einbinder et al., 2023). Moreover, calibrating using only the clean data introduces bias since the clean samples are drawn from $P_{X, Y | M=0}$, while the test distribution is $P_{X, Y}$. This bias might lead to undercoverage, as demonstrated in Figure 1.

To account for this bias, we assume the privileged information explains the corruption appearances, i.e., $(X, Y) \perp\!\!\!\perp M \mid Z$. We study the robustness of PCP (Feldman & Romano, 2024) to inaccuracies of the weights under this assumption. We then consider an alternative setup where the clean label $Y$ can be estimated well from $Z$, and develop the method UI to construct prediction sets satisfying (1) even when the weights cannot be approximated accurately. Lastly, we combine all methods into a triply robust calibration scheme which enjoys the validity guarantees of both methods. See Table 1 in Appendix C.4 for a summary of our results.

## 2 BACKGROUND AND RELATED WORK

### 2.1 CONFORMAL PREDICTION

Conformal prediction (CP) (Vovk et al., 2005) is a popular framework for generating prediction sets with a guaranteed coverage rate (1). It splits the dataset into a training set, denoted by $\mathcal{I}_1$, and a calibration set, denoted by $\mathcal{I}_2$. A learning model $\hat{f}$ is then trained on the training data and its performance is assessed on the calibration set using a non-conformity score function $\mathcal{S}(\cdot) \in \mathbb{R}$: $S_i = \mathcal{S}(X_i, Y_i; \hat{f}), \forall i \in \mathcal{I}_2$. For example, in regression problems, the score could be the absolute residual $\mathcal{S}(x, y; \hat{f}) = |\hat{f}(x) - y|$, where $\hat{f}$ represents a mean estimator. The next step is computing the $(1 + 1/|\mathcal{I}_2|)(1 - \alpha)$-th empirical quantile of the calibration scores for the nominal coverage level:

$$Q^{\text{CP}} = (1 + 1/|\mathcal{I}_2|)(1 - \alpha)\text{-th empirical quantile of the scores } \{S_i\}_{i \in \mathcal{I}_2}.$$

Finally, the prediction set for a test point is defined as:

$$C^{\text{CP}}(X^{\text{test}}) = \{y : \mathcal{S}(X^{\text{test}}, y; \hat{f}) \leq Q^{\text{CP}}\}. \tag{2}$$

This prediction set is guaranteed to achieve the desired marginal coverage rate, assuming that the calibration and test samples are exchangeable (Vovk et al., 2005). Since this is not the case in our setup, the next section introduces WCP, an extension of CP designed to handle covariate shifts.

### 2.2 WEIGHTED CONFORMAL PREDICTION

In the previous section, we noted that prediction sets constructed by CP might fail to achieve the desired coverage level due to the corruptions in the data. To overcome this issue, one could apply CP using only the scores of the uncorrupted samples, i.e., $\{\mathcal{S}(X_i, \tilde{Y}_i, \hat{f})\}_{i \in \mathcal{I}_2^{\text{uc}}}$, where $\mathcal{I}_2^{\text{uc}} = \{i \in \mathcal{I}_2 : M_i = 0\}$ are the indexes of the uncorrupted calibration samples. Although these scores are computed using the true labels and are therefore accurate, considering only the uncorrupted samples induces a covariate shift between the calibration and test data. The *weighted conformal prediction* (WCP) (Tibshirani et al., 2019) method corrects this covariate shift by weighting the

non-conformity scores using the likelihood ratio $w(z) = \frac{dP_{\text{test}}}{dP_{\text{train}}}(z)$, where $dP_{\text{test}}(z), dP_{\text{train}}(z)$ are the densities of the test and train probabilities, respectively. In this context, the weights can be expressed as $w(z) = \frac{\mathbb{P}(M=0)}{\mathbb{P}(M=0|Z=z)}$ (Feldman & Romano, 2024). Then, it extracts the quantile of the weighted distribution:

$$Q^{\text{WCP}}(Z^{\text{test}}) := \text{Quantile}\left(1-\alpha; \sum_{i \in \mathcal{I}_2^{\text{uc}}} \frac{w(Z_i)}{\Sigma_{j \in \mathcal{I}_2^{\text{uc}}} w(Z_j) + w(Z^{\text{test}})}\delta_{S_i} + \frac{w(Z^{\text{test}})}{\Sigma_{j \in \mathcal{I}_2^{\text{uc}}} w(Z_j) + w(Z^{\text{test}})}\delta_\infty\right).$$
(3)

The prediction set is constructed similarly to (2), except for using the threshold $Q^{\text{WCP}}(Z^{\text{test}})$. While this procedure is guaranteed to achieve the nominal coverage level, it cannot be applied directly since it relies on access to the test PI $Z^{\text{test}}$, which is unavailable in our framework.

### 2.3 CONFORMAL PREDICTION WITH NOISY LABELS

The works of Einbinder et al. (2023); Sesia et al. (2024); Penso & Goldberger (2024); H. Zargarbashi et al. (2024); Penso et al. (2025); Bashari et al. (2025) explore the setup where CP is employed with noisy labels. The main conclusion of Einbinder et al. (2023) is that CP remains valid label noise when the noise is dispersive, i.e., increasing the variability of the observed labels. Specifically in regression tasks, their analysis reveals that CP remains valid when the noise is symmetric and additive. Their study also explores different noise models and provides empirical evidence for the robustness of CP. Building on these results, in Section 3.2, we impute corrupted labels with noisy versions of the true ones in a way that leads to a valid coverage rate. Additional related work is given in Appendix B.

## 3 METHODS

In this section, we split our study into two cases based on the role of the PI: either as an indicator of the corruption variable $M$ or as a proxy for the label $Y$. In the first case, for example, $Z$ may represent an annotator's expertise level, such that a lower value may correspond to a higher likelihood of label noise. We present PCP, a method for constructing reliable uncertainty sets in this setup, and analyze its robustness to inaccuracies in the estimated corruption probabilities. In the second setting, we assume that the PI serves as a proxy for the label itself, for example, $Z$ may be a high-resolution image or detailed clinical reports available only during training. For this case, we develop UI—a novel imputation technique that leverages $Z$ to generate theoretically valid uncertainty sets.

### 3.1 CASE 1: WHEN THE PI EXPLAINS THE CORRUPTION INDICATOR

#### 3.1.1 PRIVILEGED CONFORMAL PREDICTION

The PCP (Feldman & Romano, 2024) procedure begins by partitioning the data into a training set, $\mathcal{I}_1$, and a calibration set, $\mathcal{I}_2$. Subsequently, a predictive model $\hat{f}$ is trained on the training set, and a non-conformity score is computed for each sample in the calibration set: $S_i = \mathcal{S}(X_i, \tilde{Y}_i; \hat{f}), \forall i \in \mathcal{I}_2$. We also compute the likelihood ratio between the calibration and test distributions to calculate the weight for each sample $i$: $w_i := \frac{\mathbb{P}(M=0)}{\mathbb{P}(M=0|Z=Z_i)}$. Next, we treat each calibration point in $\mathcal{I}_2$ as a test point and apply WCP as a subroutine using the uncorrupted calibration samples to derive the score threshold $Q(Z_i)$ for the $i$-th sample. The final test score threshold, denoted as $Q^{\text{PCP}}$, is then defined as the $(1-\beta)$-th empirical quantile of the calibration thresholds $\{Q(Z_i)\}_{\{i \in \mathcal{I}_2\}}$:

$$Q^{\text{PCP}} = \text{Quantile}\left(1-\beta; \sum_{i \in \mathcal{I}_2} \frac{1}{|\mathcal{I}_2|+1}\delta_{Q(Z_i)} + \frac{1}{|\mathcal{I}_2|+1}\delta_\infty\right),$$

where $\beta \in (0, \alpha)$ is a pre-defined level, e.g., $\beta = 0.05$. Finally, for a new test input $X^{\text{test}}$, the prediction set for $Y^{\text{test}}$ is constructed as follows: $C^{\text{PCP}}(X^{\text{test}}) = \left\{y : \mathcal{S}(X^{\text{test}}, y, \hat{f}) \leq Q^{\text{PCP}}\right\}$. This prediction set is guaranteed to obtain a valid coverage rate, as stated next.

**Theorem 1** (Validity of PCP (Feldman & Romano, 2024)). *Suppose that $\{(X_i, Y_i, \tilde{Y}_i, Z_i, M_i)\}_{i=1}^{n+1}$ are exchangeable, $Y \perp\!\!\!\perp M \mid Z$, and $P_Z$ is absolutely continuous with respect to $P_{Z|M=0}$. Then, the prediction set $C^{\text{PCP}}(X^{\text{test}})$ achieves the desired coverage rate: $\mathbb{P}(Y^{\text{test}} \in C^{\text{PCP}}(X^{\text{test}})) \geq 1-\alpha$.*

The above theorem provides a valid coverage rate guarantee even without access to the test PI $Z^{\text{test}}$, and despite the corruptions present in the data. Nevertheless, the true weights $w_i$ are required for this guarantee to hold. In the following section, we study the robustness of PCP to inaccurate weights. Surprisingly, our analysis provided hereafter reveals that PCP can achieve the nominal coverage rate even when applied with inaccurate approximates of $w_i$.

### 3.1.2 IS PCP ROBUST TO INACCURATE WEIGHTS?

The following robustness analysis of PCP is divided into two parts. First, we consider a case where the inaccurate weights $\{\tilde{w}_i\}_{i=1}^n$ are shifted by a constant error $\delta \in \mathbb{R}$ from the true weights for all samples. In the second, we extend the analysis to a more general setting where the error varies across samples. We remark that the theory developed in this section also applies to WCP, as detailed in Appendix A.1. We begin by examining the case of constant error, formulated as:

$$\tilde{w}_i := w_i + \delta, \ \forall i = 1, ..., n. \tag{4}$$

We remark that, in this analysis, we do not consider the sign of the weights and allow them to become negative. We denote the sum of the true weights by $W_k := \sum_{j=1}^k w_j$ and recall that $Q^{\text{WCP}}$ from (3) is the threshold constructed by WCP using the true weights. We also denote by $Q^{\text{CP}}$ the threshold generated by Naive CP, which is CP from Section 2.1 applied using only the uncorrupted data; see Appendix A.1.1 for more details. These notations set the ground for the conditions required for PCP to achieve the desired coverage rate.

**Theorem 2.** *Suppose that the assumptions of Theorem 1 hold. Further, suppose that at least one of the following holds: (1)* $\mathbb{P}\left(Q^{CP} > Q^{WCP}\right) \geq 1-\varepsilon$ *and* $\delta \geq 0$; *(2)* $\mathbb{P}\left(Q^{CP} > Q^{WCP}, \delta < -\frac{W_{n+1}}{n+1}\right) \geq 1-\varepsilon$; *(3)* $\mathbb{P}\left(Q^{CP} < Q^{WCP}, \delta > -\frac{W_{n+1}}{n+1}\right) \geq 1 - \varepsilon$ *and* $\delta \leq 0$; *(4)* $\mathbb{P}\left(Q^{CP} = Q^{WCP}\right) \geq 1 - \varepsilon$. *Then, the prediction set* $C^{PCP}(X^{test})$ *constructed by PCP with weights shifted by* $\delta$, *as in* (4), *satisfies:*

$$\mathbb{P}(Y^{test} \in C^{PCP}(X^{test})) \geq 1 - \alpha - \varepsilon.$$

The proofs are provided in Appendix A.1. If Naive CP attains the nominal coverage rate, i.e., $Q^{\text{CP}} > Q^{\text{WCP}}$, PCP also achieves high coverage even when the weights are poorly estimated ($\delta \geq 0$ or $\delta$ is sufficiently negative). However, if Naive CP undercovers, i.e., $Q^{\text{CP}} < Q^{\text{WCP}}$, then the weights must be accurate for PCP to be valid, specifically, $\delta$ must lie within the narrow interval $\left(-\frac{W_{n+1}}{n+1}, 0\right)$.

We demonstrate Theorem 2 on two synthetic datasets. In the first dataset, Naive CP achieves over-coverage, while in the second, Naive CP undercovers the response. For each dataset, we apply PCP using weights shifted by $\delta$, as in (4). We refer to Appendix D.3 for the full details about this experimental setup. The coverage rates of PCP for different values of $\delta$ are shown in Figure 2. This figure indicates that when Naive CP over-covers, PCP achieves valid coverage for $\delta \geq 0$ or $\delta < -\frac{W_{n+1}}{n+1} \approx -1$. However, when Naive CP undercovers, $\delta$ must lie within $\left(-\frac{W_{n+1}}{n+1}, 0\right)$ for PCP to achieve the nominal coverage rate. This result is connected to the MEPS experiment from Figure 1 in which Naive CP and PCP undercover the response, indicating that the weight error does not fall inside the interval. In conclusion, the empirical regions of $\delta$ in which PCP is valid that we observed in this experiment align with the theoretical bounds from Theorem 2.

Next, we consider the setting in which the errors are not uniform:

$$\tilde{w}_i = w_i + \delta_i, \ \forall i = 1, ..., n. \tag{5}$$

We assume that the errors are bounded: $\delta_i \in [\delta_{\min}, \delta_{\max}]$ for some $\delta_{\min}, \delta_{\max} \in \mathbb{R}$ and $\tilde{\delta}_i = (\delta_i - \delta_{\min})/(\delta_{\max} - \delta_{\min})$ are the normalized errors. We denote the XOR operator by $(a \text{ XOR } b) = ((\text{not } a) \text{ and } b) \text{ or } (a \text{ and } (\text{not } b))$, and the NXOR operator by $(a \text{ NXOR } b) = (\text{not } (a \text{ XOR } b))$. We also denote by $k^{\text{WCP}}$ the index of the score corresponding to the threshold $Q^{\text{WCP}}$ from (3), generated by WCP with true weights: $k^{\text{WCP}} := \min\left\{k : \sum_{i=1}^k w_i/W_{n+1} \geq 1 - \alpha\right\}$. We now derive the conditions under which PCP applied with weights of a general error achieves valid coverage.

**Theorem 3.** *Suppose that the assumptions in Theorem 1 hold. Further suppose that one of the following is satisfied:*

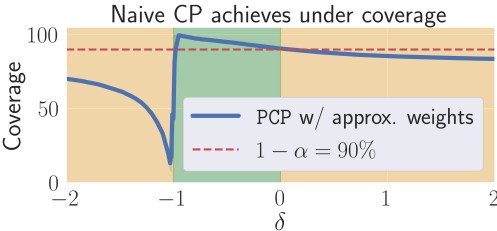 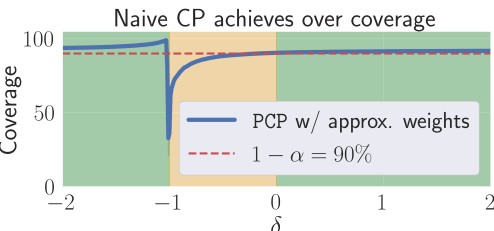

Figure 2: The coverage rate of `PCP` applied on synthetic data with weights shifted by $\delta$. Green: valid coverage region, orange: invalid coverage region. Left: `Naive CP` under-covers the response. Right: `Naive CP` achieves over-coverage. Results are averaged over 20 random splits of the data.

1. $\mathbb{P}\left(Q^{CP} < Q^{WCP}, \left(\bar{\bar{\delta}} \leq \frac{\tilde{\Delta}_{n+1}W_{k^{WCP}} - \tilde{\Delta}_{k^{WCP}}W_{n+1}}{W_{n+1}k^{WCP} - (n+1)W_{k^{WCP}}} \; NXOR \; \delta_{min} > -\frac{\delta_{max}\tilde{\Delta}_{n+1} + W_{n+1}}{n+1-\tilde{\Delta}_{n+1}}\right)\right) \geq 1 - \varepsilon,$

2. $\mathbb{P}\left(Q^{CP} > Q^{WCP}, \left(\bar{\bar{\delta}} \leq \frac{\tilde{\Delta}_{n+1}W_{k^{WCP}} - \tilde{\Delta}_{k^{WCP}}W_{n+1}}{W_{n+1}k^{WCP} - (n+1)W_{k^{WCP}}} \; XOR \; \delta_{min} > -\frac{\delta_{max}\tilde{\Delta}_{n+1} + W_{n+1}}{n+1-\tilde{\Delta}_{n+1}}\right)\right) \geq 1 - \varepsilon,$

3. $\mathbb{P}\left(Q^{CP} = Q^{WCP}, \left(\bar{\bar{\delta}} \leq \frac{\tilde{\Delta}_{n+1}W_{k^{WCP}} - \tilde{\Delta}_{k^{WCP}}W_{n+1}}{W_{n+1}k^{WCP} - (n+1)W_{k^{WCP}}} \; NXOR \; \frac{\tilde{\Delta}_{k^{WCP}}}{\tilde{\Delta}_{n+1}} \leq \frac{W_{k^{WCP}}}{C_{n+1}}\right)\right) \geq 1 - \varepsilon.$

*Above, $\tilde{\Delta}_k := \sum_{i=1}^{k} \tilde{\delta}_i$, and $\bar{\bar{\delta}} := \delta_{min}/(\delta_{max} - \delta_{min})$. Then, the prediction set $C^{PCP}(X^{test})$ constructed by `PCP` with weights shifted by a general error, as in (5), attains the following coverage rate:*

$$\mathbb{P}(Y^{test} \in C^{PCP}(X^{test})) \geq 1 - \alpha - \varepsilon.$$

Theorem 3 establishes the connection between the errors $\delta_i$ and the validity of `PCP`. Similarly to the constant error setting, the validity of `PCP` depends on whether `Naive CP` obtains overcoverage or undercoverage. Here, however, the region of $\delta_{min}, \delta_{max}$ in which we attain valid coverage is more complex and determined by the true weights $w_i$ and the distribution of the normalized errors $\tilde{\delta}_i$. In the case that `Naive CP` achieves overcoverage, the validity region is defined by the XOR of two variables, whereas in the case of undercoverage, the validity region is the exact complement, and given by the NXOR of the same variables.

Figure 8 in Appendix F.2 illustrates the coverage validity of `PCP` along with the theoretical bounds derived in Theorem 3 for various combinations of $\delta_{min}, \delta_{min}$. We experiment on the same synthetic datasets from Figure 2 while using weights with varying errors. We examine two distributions of $\delta_i$: uniform distribution and right-skewed distribution, where most samples are concentrated in the top 5% of the range $(\delta_{min}, \delta_{max})$. See Appendix E.2 for further details regarding the experimental setup. This figure shows that the bounds of Theorem 3 align with the empirical coverage validity of `PCP`, with minor discrepancies arising from interpolating discrete values. Furthermore, this figure reveals that different error distributions of $\delta_i$ yield diverse validity regions: the validity region of a uniform distribution is diagonal, while the validity region of the right-skewed distribution is horizontal. Moreover, we observe a pattern similar to the one observed in Figure 2: when `Naive CP` overcovers the response, the validity region is extensive, spanning almost the entire space, excluding one interval. In contrast, when `Naive CP` undercovers, the validity region is limited to one interval. Finally, we refer to Appendix F.2 for additional experiments demonstrating the effect of inaccurate weights on the coverage rate attained by `PCP`.

### 3.2 CASE 2: WHEN THE PI EXPLAINS THE LABEL

In this section, we introduce *uncertain imputation* (`UI`), a novel and different approach to address corrupted labels, which, in contrast to `PCP`, does not require access to the conditional corruption probabilities. With `UI`, however, we use the PI, which is always observed, to impute the corrupted labels with an uncertain version of them, and show that this procedure achieves a valid coverage rate under several assumptions. This way, `UI` can obtain valid coverage even when `PCP` does not. Intuitively, `UI` is more useful than `PCP` when $Y$ is relatively easy to predict given $X, Z$.

We begin by splitting the data into three parts: a training set, $\mathcal{I}_1$, a calibration set, $\mathcal{I}_2$, and a reference set $\mathcal{I}_3$, from which the residual errors will be sampled to account for the uncertainty in the estimated

labels. Then, we fit two predictive models to estimate the response using the training data: (1) a predictive model $\hat{f}(x)$ that takes as an input only the feature vector $X$ – as in standard CP; and (2) an additional predictive model $\hat{g}(x,z)$ that utilizes both the feature vector $X$ and the PI $Z$ to predict $Y$. Next, we compute the residual error of $\hat{g}$ for each point in the reference set:

$$E_i = Y_i - \hat{g}(X_i, Z_i), \forall i \in \mathcal{I}_3. \tag{6}$$

We define $\mathcal{E}(z)$ as a reference set of holdout errors conditional on $z$, i.e., $\mathcal{E}(z) := \{E_i : i \in \mathcal{I}_3, Z_i = z, M_i = 0\}$ and denote by $E(z)$ a random variable drawn uniformly from $\mathcal{E}(z)$. We impute the corrupted labels using this set, and denote the imputed label by $\bar{Y}_i$:

$$\bar{Y}_i = \begin{cases} Y_i & \text{if } M_i = 0, \\ \hat{g}(X_i, Z_i) + E(Z_i) & \text{otherwise} \end{cases}, \forall i \in \mathcal{I}_2 \tag{7}$$

Next, we compute the non-conformity scores of the imputed calibration set, denoted by $\bar{S}_i$: $\bar{S}_i = \mathcal{S}(X_i, \bar{Y}_i; \hat{f}), \forall i \in \mathcal{I}_2$, and define the threshold using these scores:

$$Q^{\text{UI}} := \text{Quantile}\left(1 - \alpha; \sum_{i \in \mathcal{I}_2} \frac{1}{|\mathcal{I}_2| + 1}\delta_{\bar{S}_i} + \frac{1}{|\mathcal{I}_2| + 1}\delta_{\infty}\right).$$

Finally, for a new input data $X^{\text{test}}$, we construct the prediction set for $Y^{\text{test}}$ as: $C^{\text{UI}}(X^{\text{test}}) = \{y : \mathcal{S}(X^{\text{test}}, y, \hat{f}) \leq Q^{\text{UI}}\}$. We summarize this procedure in Algorithm 2 in Appendix C.3. We now show that UI achieves a valid marginal coverage rate if (1) $\hat{g}$ is sufficiently accurate; and (2) the set $C^{\text{UI}}$ contains all peaks of the distribution of $Y \mid X, Z$, as formulated next.

**Theorem 4.** *Suppose that $\{(X_i, Y_i, \tilde{Y}_i, Z_i, M_i)\}_{i=1}^{n+1}$ are i.i.d., and $(X, Y) \perp\!\!\!\perp M \mid Z$. Denote by $C^{\text{UI}}(x) = [a(x), b(x)]$ the prediction set constructed by UI that draws errors from the true distribution of $E \mid Z$. Suppose that $Y$ follows the model: $Y = g^*(X, Z) + \varepsilon$, where $\varepsilon$ is drawn from a distribution $P_{E^*}$ and $\varepsilon \perp\!\!\!\perp X \mid Z$. Further, suppose that*

1. *$\hat{g}$ is sufficiently accurate so that there exists a residual $R^{test}$ satisfying: (a) $\hat{g}(X^{test}, Z^{test}) = g^*(X^{test}, Z^{test}) + R^{test}$, and (b) $R^{test} \perp\!\!\!\perp (g^*(X^{test}, Z^{test}), C^{\text{UI}}(X^{test})) \mid Z^{test}$.*

2. *For every $z \in \mathcal{Z}$ and $x \in \mathcal{X}$ such that $f_{X^{test}, Z^{test}}(x, z) > 0$ the density of $Y^{test} \mid X^{test} = x, Z^{test} = z$ is peaked inside the interval $[a(x), b(x)]$, i.e.,*

    *(a) $\forall v > 0 : f_{Y^{test} \mid X^{test} = x, Z^{test} = z}(b(x) + v) \leq f_{Y^{test} \mid X^{test} = x, Z^{test} = z}(b(x) - v)$, and*

    *(b) $\forall v > 0 : f_{Y^{test} \mid X^{test} = x, Z^{test} = z}(a(x) - v) \leq f_{Y^{test} \mid X^{test} = x, Z^{test} = z}(a(x) + v)$.*

*Then, $\mathbb{P}(Y^{test} \in C^{\text{UI}}(X^{test})) \geq 1 - \alpha$.*

The proof is given in Appendix A.2. We pause here to clarify the assumptions of Theorem 4. The first assumption requires that the residual errors are independent of the prediction of $\hat{g}$ and of $C^{\text{UI}}$ given the PI $Z$. Notably, this assumption does not restrict the distribution of the errors or their magnitude and holds even under substantial errors. Intuitively, this means that the PI serves as a good proxy for $Y$. For example, in medical imaging, a pathologist's diagnostic report can act as privileged information that predicts tissue diagnosis. In this case, we require that the predictor that estimates $Y$ from the PI is accurate up to an error that is independent of $g^*, C$ conditional on $Z$. The second assumption of Theorem 4 states that the distribution of $Y \mid X, Z$ is concentrated in the predicted interval, and that the density at $Y = y$ decreases as we move away from this interval. Importantly, this assumption does not limit the number of peaks in the distribution of $Y \mid X, Z$, as long as all such peaks are inside the interval. Moreover, since the PI is strongly indicative of the label in our framework, the distribution of $Y \mid X, Z$ is expected to be relatively simple in practice. Overall, this assumption is relatively mild, particularly for prediction intervals that aim to achieve a high coverage rate. In Appendix F.5 demonstrate in experiments that UI can obtain valid coverage even when the independence requirements of this theorem are not exactly satisfied. We remark that in practical applications, if $Z$ is continuous or high-dimensional, the reference set $\mathcal{E}(z)$ might be empty or too small. To alleviate this, we recommend employing the clustering techniques described in Appendix C.3.1.

### 3.3 TRIPLY ROBUST CONFORMAL PREDICTION WITH PRIVILEGED INFORMATION

Recall that `PCP` and `UI` rely on different sets of assumptions to provide a theoretical guarantee. Therefore, we propose combining these calibration schemes to enjoy the robustness guarantees of both methods. In addition, since `Naive CP` generates valid prediction sets when the underlying model $\hat{f}$ is ideal, we include it in the ensemble of the calibration schemes. In sum, this `TriplyRobust` method takes the test features vector $X^{\text{test}}$ and unifies the prediction sets of all three methods:

$$C^{\texttt{TriplyRobust}}(X^{\text{test}}) = C^{\texttt{Naive CP}}(X^{\text{test}}) \cup C^{\texttt{PCP}}(X^{\text{test}}) \cup C^{\texttt{UI}}(X^{\text{test}}). \tag{8}$$

This approach achieves the nominal coverage rate if the assumptions of one of the methods hold:

**Theorem 5.** *Suppose that $\{(X_i, Y_i, \tilde{Y}_i, Z_i, M_i)\}_{i=1}^{n+1}$ are exchangeable, and $(X, Y) \perp\!\!\!\perp M \mid Z$. Further, suppose that at least one of the following is satisfied: (1) The model $\hat{f}$ is sufficiently accurate so that the scores $\{\mathcal{S}(X_i, \tilde{Y}_i; \hat{f}) : i = 1, ..., n, M_i = 0\} \cup \{\mathcal{S}(X^{test}, Y^{test}; \hat{f})\}$ are exchangeable; (2) The assumptions of Theorem 1 hold; (3) The assumptions of Theorem 4 hold. Then, the prediction set $C^{TriplyRobust}(X^{\text{test}})$ from (8) achieves the desired coverage rate:*

$$\mathbb{P}(Y^{\text{test}} \in C^{TriplyRobust}(X^{\text{test}})) \geq 1 - \alpha.$$

Intuitively, Theorem 5 guarantees that `TriplyRobust` generates valid uncertainty sets if at least one of the following distributions is well-estimated: $Y \mid X$ (`Naive CP`), $M \mid Z$ (`PCP`), or $Y \mid Z$ (`UI`). In the following section, we demonstrate the robustness of this approach.

## 4 EXPERIMENTS

In this section, we quantify the effectiveness of our proposed techniques through three experiments. In all experiments, the dataset is randomly split into four distinct subsets: training, validation, calibration, and test sets. For the `UI` method, the original calibration set is further partitioned into a reference set and a calibration set. The training set is used for fitting a predictive model, and the validation set is used for early stopping. The model is then calibrated using the calibration data, and the performance is evaluated on the test set over 30 random data splits. Moreover, we use the CQR (Romano et al., 2019) non-conformity scores, aiming to obtain $1 - \alpha = 90\%$ coverage rate. The full details regarding the training methodology, datasets, corruption techniques, and the overall experimental protocol are given in Appendix E. We further demonstrate our proposals on a causal inference task using the NSLM dataset (Yeager et al., 2019) in Appendix F.3.

### 4.1 SYNTHETIC EXPERIMENT: THE ROBUSTNESS OF TRIPLYROBUST

We demonstrate the robustness of `TriplyRobust` by combining degenerate and oracle variants of each underlying component: quantile regression (QR), `PCP`, and `UI`. We generate a synthetic dataset with 10-dimensional inputs $X$, 3-dimensional privileged information $Z$, and a continuous response $Y$ that is a function of $X$ and $Z$. The dataset is artificially corrupted by removing labels, such that labels with greater uncertainty have a higher probability of being removed. The degenerate variant of QR outputs the trivial prediction set $\{0\}$, while the oracle version uses the true conditional quantiles of $Y \mid X$ to achieve the desired conditional coverage. Similarly, the degenerate version of `PCP` computes weights using half the true corruption probabilities. In contrast, its oracle counterpart employs the correct probabilities. The trivial imputation method assigns missing labels the value 0, while the oracle method draws labels from the true conditional distribution of $Y \mid Z$. Figure 3 shows that when all three components are degenerate, `TriplyRobust` achieves a coverage rate lower than the nominal level. However, when at least one technique is oracle-based, `TriplyRobust` produces valid uncertainty estimates. This experiment reveals that the coverage rate achieved by `TriplyRobust` is not overly conservative, despite being a union of three intervals.

### 4.2 SYNTHETIC EXPERIMENT: WEIGHTS ARE HARD TO ESTIMATE

To demonstrate the advantages of `UI` compared to `PCP`, we adopt the synthetic setup from Feldman & Romano (2024), where $Z$ is a strong predictor of $Y$. The only difference is that we engineer the missingness mechanism to be challenging to estimate; see Appendix D.3 for details. Figure 4

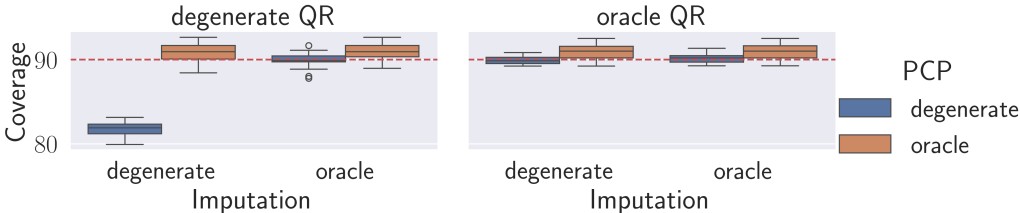

Figure 3: The coverage achieved by `TriplyRobust` with "degenerate" or "oracle" models.

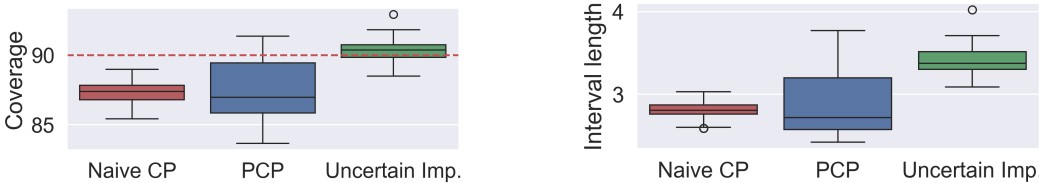

Figure 4: **Complex weights experiment.** The performance of `Naive CP`, `PCP` and `UI`.

shows that `PCP` fails to attain the nominal 90% coverage rate due to inaccuracies in the estimated weights. In contrast, `UI` relies on accurate estimates of $Y$ from $(X, Z)$ rather than on weights; therefore, it consistently achieves the desired coverage, as guaranteed by Theorem 4. This experiment demonstrates that `UI` can achieve the desired coverage rate, even in cases where `PCP` does not.

### 4.3 REAL DATASETS WITH ARTIFICIAL CORRUPTIONS

In this section, we evaluate the proposed `UI` in a missing response setup using five benchmarks used in Feldman & Romano (2024): Facebook1,2 (facebook), Bio (bio), House (house), Meps19 (meps_19). We follow Feldman & Romano (2024) and artificially define the PI as the feature in $X$ with the highest correlation to $Y$ and remove it from $X$, so that the PI is unavailable at test time. Since all true labels are available in the original datasets, we artificially remove 20% of them similarly to Feldman & Romano (2024) to induce a distribution shift between missing and observed variables.

We employ the naive conformal prediction (`Naive CP`) that uses only the observed labels, `PCP`, with either estimated corruption probabilities or true ones used for computing the weights $w(z)$, and the proposed `UI`. Additionally, to demonstrate the importance of the error sampling scheme of `UI`, we apply `CP` with a naive imputation scheme that replaces missing labels with the mean estimates of $Y \mid X, Z$ (`Naive Imputation`). The performances of all calibration schemes are presented in Figure 5. This figure indicates that the naive approaches produce too narrow intervals that do not achieve the desired coverage level. This is anticipated, as the validity guarantees of `CP` do not hold under distribution shifts or naive imputations. However, `PCP`, and the proposed `UI` consistently achieve the target 90% coverage level, as they appropriately account for the distributional shift. This is also indicated by Theorem 4. Notably, since `PCP` with estimated weights is valid while `Naive CP` is not, Figure 2 suggests that weight estimation errors fall within the theoretical validity region. Overall, this experiment reveals that `UI` constructs uncertainty intervals that are both statistically efficient and reliable. The performance of `TriplyRobust` in this experiment is provided in Appendix F.6.

## 5 DISCUSSION AND IMPACT STATEMENT

In this work, we analyzed the impact of inaccurate weights on `WCP` and `PCP`, and introduced `UI`, a novel calibration technique for reliable uncertainty quantification under corrupted labels. Our theoretical guarantees, supported by empirical experiments, demonstrate that `UI` constructs valid uncertainty estimates. While the validity conditions we derived for `WCP` and `PCP` are theoretically grounded, they require access to true weights, which are unavailable in practice. This calls for a promising future research direction in estimating these conditions from the available data. One limitation of our proposed `UI` is that it requires the features and responses to be independent of

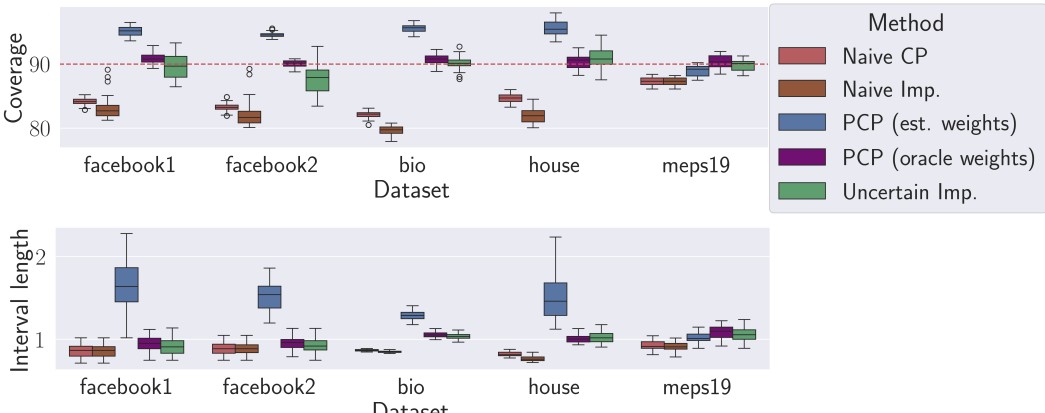

Figure 5: **Missing response experiment.** The performance of various methods; see text for details.

the corruption indicator given the privileged information, which resembles the strong ignorability assumption in causal inference (Rubin, 1978; Rosenbaum & Rubin, 1983; Imbens & Rubin, 2015). Furthermore, our approach assumes that the label variability depends only on the PI and that the labels can be accurately estimated from the features and PI. In practice, our experiments indicate that our method attains valid coverage. Future work could explore extending our theoretical guarantees to multiple-annotator settings and incorporating privileged information into ambiguity-aware calibration methods Javanmardi et al. (2024); Caprio et al. (2025). Finally, we acknowledge potential social implications, akin to many developments in ML.

ACKNOWLEDGMENTS

Funded by the European Union (ERC, SafetyBounds, 101163414). Views and opinions expressed are however those of the authors only and do not necessarily reflect those of the European Union or European Research Council Executive Agency (ERCEA). Neither the European Union nor the granting authority can be held responsible for them. Y.R. thanks the Career Advancement Fellowship, Technion.

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

# A    THEORETICAL RESULTS

## A.1    ANALYSIS OF INACCURATE WEIGHTS ON WEIGHTED CONFORMAL PREDICTION

In this section, we analyze the coverage rate achieved by WCP applied with estimated (inaccurate) weights. Before we present the analysis, we begin by defining notations that will be used throughout the theoretical analysis.

### A.1.1    NOTATIONS

Suppose, without loss of generality, that the indices of the calibration set are $\{1, \ldots, n\}$. For the simplicity of the proof, we suppose without loss of generality that the indices are sorted by the score, i.e.,
$$\forall i \in \{1, \ldots, n-1\} : S_i \leq S_{i+1}$$
For ease of notations, we define $S_{n+1} = \infty$, while the score of the test sample is denoted by $\mathcal{S}(X_{n+1}, Y_{n+1}; \hat{f})$. Recall that the ground truth weights used by WCP are formulated by:
$$w_i = w(Z_i) = \frac{f_Z^{\text{test}}(Z_i)}{f_Z^{\text{train}}(Z_i)}.$$

The inaccurate weights are denoted by $\hat{w}_i$. The true normalized weights are formulated by: $p_i = \frac{w_i}{W_{n+1}}$, where $W_k := \sum_{j=1}^{k} w_j$ is a partial sum of weights. Similarly, we denote the normalized inaccurate weights by $\hat{p}_i$. We denote the threshold chosen by WCP with the oracle weights $\{w_i\}_{i=1}^{n+1}$ by $Q^{\text{WCP}}$, which is the $(1-\alpha)$ empirical quantile of the distribution $\sum_{i=1}^{n} p_i \delta_{S_i} + p_{n+1} \delta_{\infty}$. The corresponding index $k^{\text{WCP}}$ is defined as:

$$k^{\text{WCP}} = \min \left\{ k : \sum_{i=1}^{k} p_i \geq 1 - \alpha \right\}. \tag{9}$$

Similarly, denote by $\hat{Q}^{\text{WCP}}$ the threshold chosen by WCP applied with the inaccurate weights $\hat{w}_i$, which is the $1 - \alpha$ empirical quantile of the distribution $\sum_{i=1}^{n} \hat{p}_i \delta_{S_i} + \hat{p}_{n+1} \delta_{\infty}$. Finally, we denote by $Q^{\text{CP}}$ the threshold of naive CP applied with no weights, and by $k^{\text{CP}}$ its corresponding index:

$$k^{\text{CP}} = \min \left\{ k : \sum_{i=1}^{k} \frac{1}{n+1} \geq 1 - \alpha \right\}. \tag{10}$$

Notice that the interval constructed by WCP with oracle weights is given by:

$$C^{\text{WCP}}(x) := \{y \in \mathcal{Y} : \mathcal{S}(x, y; \hat{f}) \leq Q^{\text{WCP}}\},$$

and similarly, the interval constructed by WCP with the inaccurate weights is

$$\hat{C}^{\text{WCP}}(x) := \{y \in \mathcal{Y} : \mathcal{S}(x, y; \hat{f}) \leq \hat{Q}^{\text{WCP}}\}. \tag{11}$$

### A.1.2    CONSTANT ERROR

In this section, we consider the setting where the inaccurate weights have a constant bias from the ground truth ones across all samples. Formally, we assume that there exists $\delta \in \mathbb{R}$ such that:
$$\hat{w}_i := w_i + \delta, \forall i \in \{1, \ldots, n\}.$$
The normalized inaccurate weights are therefore given by:
$$\hat{p}_i := \frac{\hat{w}_i}{\sum_{j=1}^{n+1} \hat{w}_j} = \frac{w_i + \delta}{\sum_{j=1}^{n+1} (w_j + \delta)} = \frac{w_i + \delta}{W_{n+1} + (n+1)\delta}.$$

We begin with a general lemma that provides a deterministic connection between $k^{\text{CP}}$ and $k^{\text{WCP}}$, as defined in (10) and (9), respectively.

**Lemma 1.** *Suppose that the calibration set is fixed, and equals to $\{(X_i, Y_i, Z_i) = (x_i, y_i, z_i)\}_{i=1}^{n}$ for some $x_i \in \mathcal{X}, y_i \in \mathcal{Y}, z_i \in \mathcal{Z}$ for all $i \in \{1, \ldots, n\}$. Further suppose that the test PI is fixed as well, i.e., $Z_{n+1} = z$ for some $z \in \mathcal{Z}$. Then, $\hat{Q}^{\text{WCP}} \geq Q^{\text{WCP}}$ if and only if one of the following is satisfied:*

1. $k^{CP} > k^{WCP}$ and ($\delta \geq 0$ or $\delta < -\frac{W_{n+1}}{n+1}$)

2. $k^{CP} < k^{WCP}$ and $-\frac{W_{n+1}}{n+1} < \delta \leq 0$

3. $k^{CP} = k^{WCP}$

*Furthermore, $\hat{Q}^{WCP} \geq Q^{CP}$ if and only if one of the following is satisfied:*

1. $k^{CP} > k^{WCP}$ and ($\delta < -\frac{W_{n+1}}{n+1}$)

2. $k^{CP} < k^{WCP}$ and $-\frac{W_{n+1}}{n+1} < \delta$

3. $k^{CP} = k^{WCP}$

*Proof.* If $\delta = 0$ then $\hat{w}_i = w_i$ for all $i$ and thus the intervals constructed by WCP with the oracle weights $w_i$ are identical to the ones constructed with the inaccurate weights $\hat{w}_i$ and therefore achieve the same coverage rate. From this point on, we only consider $\delta \neq 0$. Furthermore, if $\delta = -\frac{W_{n+1}}{n+1}$ then $\hat{p}_i$ is not defined and hence we do not consider this case and suppose that $\delta \neq -\frac{W_{n+1}}{n+1}$. Observe that

$$\sum_{i=1}^{k^{WCP}} \hat{p}_i \leq \sum_{i=1}^{k^{WCP}} p_i \iff \hat{k}^{WCP} \geq k^{WCP} \iff \hat{Q}^{WCP} \geq Q^{WCP}.$$

We now analyze when $\sum_{i=1}^{k^{WCP}} \hat{p}_i \leq \sum_{i=1}^{k^{WCP}} p_i$ is satisfied. We split into three cases. First, we consider $\delta > 0$:

$$\sum_{i=1}^{k^{WCP}} \hat{p}_i \leq \sum_{i=1}^{k^{WCP}} p_i$$
$$\frac{W_{k^{WCP}} + k^{WCP}\delta}{W_{n+1} + (n+1)\delta} \leq \frac{W_{k^{WCP}}}{W_{n+1}}$$
$$(W_{k^{WCP}} + k^{WCP}\delta)W_{n+1} \leq W_{k^{WCP}}(W_{n+1} + (n+1)\delta)$$
$$k^{WCP}\delta W_{n+1} \leq W_{k^{WCP}}(n+1)\delta$$
$$\frac{k^{WCP}}{n+1}\delta \leq \frac{W_{k^{WCP}}}{W_{n+1}}\delta$$
$$\frac{k^{WCP}}{n+1} \leq \frac{W_{k^{WCP}}}{W_{n+1}}$$
$$k^{CP} \geq k^{WCP}.$$

We now turn to $-\frac{W_{n+1}}{n+1} < \delta < 0$:

$$\sum_{i=1}^{k^{WCP}} \hat{p}_i \leq \sum_{i=1}^{k^{WCP}} p_i$$
$$\frac{W_{k^{WCP}} + k^{WCP}\delta}{W_{n+1} + (n+1)\delta} \leq \frac{W_{k^{WCP}}}{W_{n+1}}$$
$$(W_{k^{WCP}} + k^{WCP}\delta)W_{n+1} \leq W_{k^{WCP}}(W_{n+1} + (n+1)\delta)$$
$$k^{WCP}\delta W_{n+1} \leq W_{k^{WCP}}(n+1)\delta$$
$$\frac{k^{WCP}}{n+1}\delta \leq \frac{W_{k^{WCP}}}{W_{n+1}}\delta$$
$$\frac{k^{WCP}}{n+1} \geq \frac{W_{k^{WCP}}}{W_{n+1}}$$
$$k^{CP} \leq k^{WCP}.$$

Finally, if $\delta < -\frac{W_{n+1}}{n+1}$:

$$\sum_{i=1}^{k^{\text{WCP}}} \hat{p}_i \leq \sum_{i=1}^{k^{\text{WCP}}} p_i$$

$$\frac{W_{k^{\text{WCP}}} + k^{\text{WCP}}\delta}{W_{n+1} + (n+1)\delta} \leq \frac{W_{k^{\text{WCP}}}}{W_{n+1}}$$

$$(W_{k^{\text{WCP}}} + k^{\text{WCP}}\delta)W_{n+1} \geq W_{k^{\text{WCP}}}(W_{n+1} + (n+1)\delta)$$

$$k^{\text{WCP}}\delta W_{n+1} \geq W_{k^{\text{WCP}}}(n+1)\delta$$

$$\frac{k^{\text{WCP}}}{n+1}\delta \geq \frac{W_{k^{\text{WCP}}}}{W_{n+1}}\delta$$

$$\frac{k^{\text{WCP}}}{n+1} \leq \frac{W_{k^{\text{WCP}}}}{W_{n+1}}$$

$$k^{\text{CP}} \geq k^{\text{WCP}}$$

We now compare the threshold of WCP applied with inaccurate weights to the threshold of naive CP:

$$\hat{Q}^{\text{WCP}} \geq Q^{\text{CP}} \iff \hat{k}^{\text{WCP}} \geq k^{\text{CP}} \iff \sum_{i=1}^{k^{\text{CP}}} \hat{p}_i \leq \sum_{i=1}^{k^{\text{CP}}} \frac{1}{n+1} \iff \frac{W_{k^{\text{CP}}} + k^{\text{CP}}\delta}{W_{n+1} + (n+1)\delta} \leq \frac{k^{\text{CP}}}{n+1}.$$

If $\delta > -\frac{W_{n+1}}{n+1}$ then:

$$\frac{W_{k^{\text{CP}}} + k^{\text{CP}}\delta}{W_{n+1} + (n+1)\delta} \leq \frac{k^{\text{CP}}}{n+1}$$

$$(W_{k^{\text{CP}}} + k^{\text{CP}}\delta)(n+1) \leq k^{\text{CP}}W_{n+1} + k^{\text{CP}}(n+1)\delta$$

$$W_{k^{\text{CP}}}(n+1) \leq k^{\text{CP}}W_{n+1}$$

$$W_{k^{\text{CP}}}(n+1) \leq k^{\text{CP}}W_{n+1}$$

$$\frac{W_{k^{\text{CP}}}}{W_{n+1}} \leq \frac{k}{n+1}$$

$$k^{\text{CP}} \leq k^{\text{WCP}}.$$

If $\delta < -\frac{W_{n+1}}{n+1}$ then:

$$\hat{Q}^{\text{WCP}} \geq Q^{\text{CP}} \iff \frac{W_{k^{\text{CP}}}}{W_{n+1}} \geq \frac{k^{\text{CP}}}{n+1} \iff k^{\text{CP}} \geq k^{\text{WCP}}.$$

$\square$

We now present a stochastic result that states the conditions under which WCP achieves a conservative coverage rate with inaccurate weights. Here, the randomness is taken over random draws of $(X_{n+1}, Y_{n+1})$ from $P_{X,Y|Z}$.

**Proposition 1.** *Suppose that the calibration set is fixed, and equals to $\{(X_i, Y_i, Z_i) = (x_i, y_i, z_i)\}_{i=1}^n$ for some $x_i \in \mathcal{X}, y_i \in \mathcal{Y}, z_i \in \mathcal{Z}$ for all $i \in \{1, \ldots, n\}$. Further suppose that the test PI is fixed as well, i.e., $Z_{n+1} = z$ for some $z \in \mathcal{Z}$. Then, $\hat{C}^{\text{WCP}}(X_{n+1})$, as defined in (11), achieves a conservative coverage rate, i.e.:*

$$\mathbb{P}(Y_{n+1} \in \hat{C}^{\text{WCP}}(X_{n+1}) \mid Z_{n+1} = z, \{(X_i, Y_i, Z_i) = (x_i, y_i, z_i)\}_{i=1}^n)$$
$$\geq \mathbb{P}(Y_{n+1} \in C^{\text{WCP}}(X_{n+1}) \mid Z_{n+1} = z, \{(X_i, Y_i, Z_i) = (x_i, y_i, z_i)\}_{i=1}^n)$$

*if one of the following is satisfied:*

1. *$k^{\text{CP}} > k^{\text{WCP}}$ and ($\delta \geq 0$ or $\delta < -\frac{W_{n+1}}{n+1}$)*

2. *$k^{\text{CP}} < k^{\text{WCP}}$ and $-\frac{W_{n+1}}{n+1} < \delta \leq 0$*

    3. $k^{CP} = k^{WCP}$.

*If the above holds with probability at least $1 - \varepsilon$ over the drawing of $Z_{n+1}$, then we get a high marginal coverage rate:*

$$\mathbb{P}(Y_{n+1} \in \hat{C}^{WCP}(X_{n+1}) \mid \{(X_i, Y_i, Z_i) = (x_i, y_i, z_i)\}_{i=1}^n)$$
$$\geq \mathbb{P}(Y_{n+1} \in C^{WCP}(X_{n+1}) \mid \{(X_i, Y_i, Z_i) = (x_i, y_i, z_i)\}_{i=1}^n) - \varepsilon.$$

*Furthermore, $\hat{C}^{WCP}(X_{n+1})$ achieves a higher coverage rate than Naive CP, i.e.,*

$$\mathbb{P}(Y_{n+1} \in \hat{C}^{WCP}(X_{n+1}) \mid Z_{n+1} = z, \{(X_i, Y_i, Z_i) = (x_i, y_i, z_i)\}_{i=1}^n)$$
$$\geq \mathbb{P}(Y_{n+1} \in C^{CP}(X_{n+1}) \mid Z_{n+1} = z, \{(X_i, Y_i, Z_i) = (x_i, y_i, z_i)\}_{i=1}^n)$$

*if one of the following is satisfied*

    1. $k^{CP} > k^{WCP}$ and $\delta < -\frac{W_{n+1}}{n+1}$

    2. $k^{CP} < k^{WCP}$ and $-\frac{W_{n+1}}{n+1} < \delta$

    3. $k^{CP} = k^{WCP}$.

*If the above holds with probability at least $1 - \varepsilon$ for the drawing of $Z_{n+1}$, then, we get a high marginal coverage rate:*

$$\mathbb{P}(Y_{n+1} \in \hat{C}^{WCP}(X_{n+1}) \mid \{(X_i, Y_i, Z_i) = (x_i, y_i, z_i)\}_{i=1}^n)$$
$$\geq \mathbb{P}(Y_{n+1} \in C^{CP}(X_{n+1}) \mid \{(X_i, Y_i, Z_i) = (x_i, y_i, z_i)\}_{i=1}^n) - \varepsilon.$$

*Proof.* The probabilities in this proof are taken for drawing $(X_{n+1}, Y_{n+1}) \sim P_{X,Y}$ conditional on the fixed calibration set $\{(X_i, Y_i) = (x_i, y_i)\}_{i=1}^n$ and the fixed test PI $Z_{n+1} = z$. For ease of notation, we omit the conditioning on the calibration set in the formulas. Notice that in this case, $\hat{Q}^{WCP}, Q^{WCP}$, $Q^{CP}$ are deterministic since these are functions of the calibration set and the test PI $Z_{n+1}$. Observe that by the construction of the uncertainty sets:

$$\hat{Q}^{WCP} \geq Q^{WCP} \Rightarrow \mathbb{P}(Y_{n+1} \in \hat{C}^{WCP}(X_{n+1}) \mid Z_{n+1} = z) \geq \mathbb{P}(Y_{n+1} \in C^{WCP}(X_{n+1}) \mid Z_{n+1} = z),$$

and

$$\hat{Q}^{WCP} \geq Q^{CP} \Rightarrow \mathbb{P}(Y_{n+1} \in \hat{C}^{WCP}(X_{n+1}) \mid Z_{n+1} = z) \geq \mathbb{P}(Y_{n+1} \in C^{CP}(X_{n+1}) \mid Z_{n+1} = z).$$

Therefore, $\mathbb{P}(Y_{n+1} \in \hat{C}^{WCP}(X_{n+1}) \mid Z_{n+1} = z) \geq \mathbb{P}(Y_{n+1} \in C^{WCP}(X_{n+1}) \mid Z_{n+1} = z)$ holds if $\hat{Q}^{WCP} \geq Q^{WCP}$, which, according to Lemma 1, is equivalent to assuming that one of the following is satisfied:

    1. $k^{CP} > k^{WCP}$ and ($\delta \geq 0$ or $\delta < -\frac{W_{n+1}}{n+1}$)

    2. $k^{CP} < k^{WCP}$ and $-\frac{W_{n+1}}{n+1} < \delta \leq 0$

    3. $k^{CP} = k^{WCP}$.

Denote the event that one of the above requirements is satisfied by $E$. Following Lemma 1 we get that $\mathbb{P}(\hat{Q}^{WCP} \geq Q^{WCP} \mid E) = 1$. If $E$ holds with probability $1 - \varepsilon$, where the randomness is taken over $Z_{n+1}$, then we get:

$$\mathbb{P}(Y_{n+1} \in \hat{C}^{WCP}(X_{n+1})) \geq \mathbb{P}(Y_{n+1} \in \hat{C}^{WCP}(X_{n+1}) \mid E)\mathbb{P}(E)$$
$$\geq \mathbb{P}(Y_{n+1} \in C^{WCP}(X_{n+1}) \mid E)\mathbb{P}(E)$$
$$= \mathbb{P}(Y_{n+1} \in C^{WCP}(X_{n+1})) - \mathbb{P}(Y_{n+1} \in C^{WCP}(X_{n+1}) \mid \bar{E})\mathbb{P}(\bar{E})$$
$$\geq \mathbb{P}(Y_{n+1} \in C^{WCP}(X_{n+1})) - \varepsilon.$$

Similarly, $\mathbb{P}(Y_{n+1} \in \hat{C}^{WCP}(X_{n+1}) \mid Z_{n+1} = z) \geq \mathbb{P}(Y_{n+1} \in C^{CP}(X_{n+1}) \mid Z_{n+1} = z)$ holds if $\hat{Q}^{WCP} \geq Q^{CP}$, which, according to Lemma 1, is equivalent to assuming that one of the following is satisfied:

1. $k^{\mathrm{CP}} > k^{\mathrm{WCP}}$ and $\delta < -\frac{W_{n+1}}{n+1}$

2. $k^{\mathrm{CP}} < k^{\mathrm{WCP}}$ and $-\frac{W_{n+1}}{n+1} < \delta$

3. $k^{\mathrm{CP}} = k^{\mathrm{WCP}}$.

If the above holds with probability $1 - \varepsilon$ over random draws of $Z_{n+1} = z \in \mathcal{Z}$, then by the same reasoning as before, we get a high marginal coverage rate:

$$\mathbb{P}(Y_{n+1} \in \hat{C}^{\mathrm{WCP}}(X_{n+1})) \geq \mathbb{P}(Y_{n+1} \in C^{\mathrm{CP}}(X_{n+1})) - \varepsilon.$$

$\square$

We now present a different stochastic result in which the randomness is taken over the randomness in the drawing of all calibration and test data points.

**Proposition 2.** *Suppose that the calibration and test samples are given by $\{(X_i, Y_i, Z_i)\}_{i=1}^{n+1}$. Further, suppose that one of the following holds:*

1. $\mathbb{P}\left(k^{CP} > k^{WCP}\right) \geq 1 - \varepsilon$ *and* $\delta \geq 0$,

2. $\mathbb{P}\left(k^{CP} > k^{WCP}, \delta < -\frac{W_{n+1}}{n+1}\right) \geq 1 - \varepsilon$,

3. $\mathbb{P}\left(k^{CP} < k^{WCP}, \delta > -\frac{W_{n+1}}{n+1}\right) \geq 1 - \varepsilon$ *and* $\delta \leq 0$,

4. $\mathbb{P}\left(k^{CP} = k^{WCP}\right) \geq 1 - \varepsilon$.

*Then, we get a high marginal coverage rate:*

$$\mathbb{P}(Y_{n+1} \in \hat{C}^{WCP}(X_{n+1})) \geq \mathbb{P}(Y_{n+1} \in C^{WCP}(X_{n+1})) - \varepsilon.$$

*Proof.* Denote by $E_1$ the following event:

$$E_1 = \begin{cases} k^{\mathrm{CP}} > k^{\mathrm{WCP}}, & \text{if } \delta \geq 0, \\ k^{\mathrm{CP}} < k^{\mathrm{WCP}}, \delta > -\frac{W_{n+1}}{n+1}, & \text{if } \delta \leq 0. \end{cases}$$

We denote $E_2 = k^{\mathrm{CP}} = k^{\mathrm{WCP}}$, $E_3 = (k^{\mathrm{CP}} > k^{\mathrm{WCP}}$ and $\delta < -\frac{W_{n+1}}{n+1})$.

$$E = E_1 \text{ or } E_2 \text{ or } E_3.$$

By the construction of $E$ and our assumptions, $\mathbb{P}(E) \geq 1 - \varepsilon$. Following Lemma 1 we get that $\mathbb{P}(\hat{Q}^{\mathrm{WCP}} \geq Q^{\mathrm{WCP}} \mid E) = 1$. By combining these, we get:

$$\begin{aligned} \mathbb{P}(Y_{n+1} \in \hat{C}^{\mathrm{WCP}}(X_{n+1})) &\geq \mathbb{P}(Y_{n+1} \in \hat{C}^{\mathrm{WCP}}(X_{n+1}) \mid E)\mathbb{P}(E) \\ &\geq \mathbb{P}(Y_{n+1} \in C^{\mathrm{WCP}}(X_{n+1}) \mid E)\mathbb{P}(E) \\ &= \mathbb{P}(Y_{n+1} \in C^{\mathrm{WCP}}(X_{n+1})) - \mathbb{P}(Y_{n+1} \in C^{\mathrm{WCP}}(X_{n+1}) \mid \bar{E})\mathbb{P}(\bar{E}) \\ &\geq \mathbb{P}(Y_{n+1} \in C^{\mathrm{WCP}}(X_{n+1})) - \varepsilon. \end{aligned}$$

$\square$

Building on our theoretical results for WCP under constant weight error, we now turn to PCP applied with constant weight errors and prove Theorem 2.

*Proof of Theorem 2.* According to Feldman & Romano (2024, Theorem 1), we get that when applied with the same weights, PCP achieves a higher coverage rate than WCP, that is:

$$\mathbb{P}(Y_{n+1} \in \hat{C}^{\mathrm{PCP}}(X_{n+1})) \geq \mathbb{P}(Y_{n+1} \in \hat{C}^{\mathrm{WCP}}(X_{n+1})).$$

Since the assumptions of Proposition 2 hold, we obtain:

$$\mathbb{P}(Y_{n+1} \in \hat{C}^{\text{WCP}}(X_{n+1})) \geq \mathbb{P}(Y_{n+1} \in C^{\text{WCP}}(X_{n+1})) - \varepsilon.$$

Lastly, under the exchangeability of the samples, Tibshirani et al. (2019, Corollary 1) states that:

$$\mathbb{P}(Y_{n+1} \in C^{\text{WCP}}(X_{n+1})) \geq 1 - \alpha.$$

By combining it all, we get:

$$\mathbb{P}(Y_{n+1} \in \hat{C}^{\text{PCP}}(X_{n+1})) \geq 1 - \alpha - \varepsilon.$$

$\square$

### A.1.3 GENERAL BOUNDED ERROR

We now turn to consider the setup where the inaccurate weights $\hat{w}_i$ are at bias $\delta_i$ from the true weights $w_i$:

$$\forall i \in \{1, \ldots, n+1\} : \hat{w}_i = w_i + \delta_i.$$

The errors $\delta_i$ are assumed to be bounded by $\delta_i \in [\delta_{\min}, \delta_{\max}]$, where $\delta_{\min} < \delta_{\max} \in \mathbb{R}$. Notice that the setting where $\delta_{\min} = \delta_{\max}$ is the case analyzed in Appendix A.1.2. We denote the normalized error $\delta_i$ by: $\tilde{\delta}_i := (\delta_i - \delta_{\min})/(\delta_{\max} - \delta_{\min})$ and $\tilde{\Delta}_k = \sum_{i=1}^{k} \tilde{\delta}_i$. We also denote $\bar{\delta} = \delta_{\min}/(\delta_{\max} - \delta_{\min})$. Notice that since $\delta_{\max} > \delta_{\min}$ we get $\tilde{\Delta}_{n+1} < n+1$. We follow the notations from A.1.1 and define the following requirements:

1. $\bar{\delta} \leq \frac{\tilde{\Delta}_{n+1} W_{k^{\text{WCP}}} - \tilde{\Delta}_{k^{\text{WCP}}} W_{n+1}}{W_{n+1} k^{\text{WCP}} - (n+1) W_{k^{\text{WCP}}}}$.

2. $\delta_{\min} > -\frac{\delta_{\max} \tilde{\Delta}_{n+1} + W_{n+1}}{n+1-\tilde{\Delta}_{n+1}}$.

3. $\frac{\tilde{\Delta}_{k^{\text{WCP}}}}{\tilde{\Delta}_{n+1}} \leq \frac{W_{k^{\text{WCP}}}}{C_{n+1}}$.

We begin with a general lemma that provides a deterministic connection between $k^{\text{CP}}$ and $k^{\text{WCP}}$. For this purpose, we define

$$(a \text{ XOR } b) = ((\text{not } a) \text{ and } b) \text{ or } (a \text{ and } (\text{not } b)),$$
$$(a \text{ NXOR } b) = (\text{not } (a \text{ XOR } b)).$$

**Lemma 2.** *Suppose that the calibration set is fixed, and equals to $\{(X_i, Y_i, Z_i) = (x_i, y_i, z_i)\}_{i=1}^{n}$ for some $x_i \in \mathcal{X}, y_i \in \mathcal{Y}, z_i \in \mathcal{Z}$ for all $i \in \{1, \ldots, n\}$. Further suppose that the test PI and the errors $\{\delta_i\}_{i=1}^{n+1}$ are also fixed, i.e., $Z_{n+1} = z$ for some $z \in \mathcal{Z}$. Then, $\hat{Q}^{\text{WCP}} \geq Q^{\text{WCP}}$ if and only if one of the following is satisfied:*

1. *$k^{\text{CP}} < k^{\text{WCP}}$ and (requirement 1 NXOR requirement 2),*

2. *$k^{\text{CP}} > k^{\text{WCP}}$ and (requirement 1 XOR requirement 2),*

3. *$k^{\text{CP}} = k^{\text{WCP}}$ and (requirement 1 NXOR requirement 3).*

*Proof.* We begin the proof by developing $\sum_{i=1}^{k^{\text{WCP}}} p_i$ and $\sum_{i=1}^{k^{\text{WCP}}} \hat{p}_i$ :

$$\sum_{i=1}^{k^{\text{WCP}}} p_i = \sum_{i=1}^{k^{\text{WCP}}} \frac{w_i}{\sum_{j=1}^{n+1} w_j} = \frac{W_{k^{\text{WCP}}}}{W_{n+1}}$$

$$\sum_{i=1}^{k^{\text{WCP}}} \hat{p}_i = \sum_{i=1}^{k^{\text{WCP}}} \frac{\hat{w}_i}{\sum_{j=1}^{n+1} \hat{w}_j} = \sum_{i=1}^{k^{\text{WCP}}} \frac{w_i + \delta_i}{\sum_{j=1}^{n+1} w_j + \delta_j} = \frac{W_{k^{\text{WCP}}} + \Delta_{k^{\text{WCP}}}}{W_{n+1} + \Delta_{n+1}}.$$

Observe that:

$$\sum_{i=1}^{k^{\text{WCP}}} p_i \leq \sum_{i=1}^{k^{\text{WCP}}} \hat{p}_i \iff \hat{k}^{\text{WCP}} \geq k^{\text{WCP}} \iff \hat{Q}^{\text{WCP}} \geq Q^{\text{WCP}}.$$

We begin by developing requirement 2:

$$W_{n+1} + \Delta_{n+1} > 0$$
$$W_{n+1} + (\delta_{\max} - \delta_{\min})\tilde{\Delta}_{n+1} + (n+1)\delta_{\min} > 0$$
$$\delta_{\max}\tilde{\Delta}_{n+1} - \delta_{\min}\tilde{\Delta}_{n+1} + (n+1)\delta_{\min} > -W_{n+1}$$
$$[(n+1) - \tilde{\Delta}_{n+1}]\delta_{\min} > -\delta_{\max}\tilde{\Delta}_{n+1} - W_{n+1}$$
$$\delta_{\min} > -\frac{\delta_{\max}\tilde{\Delta}_{n+1} + W_{n+1}}{n+1-\tilde{\Delta}_{n+1}}$$

We now turn to requirement 1, considering the case where $\frac{k^{\text{WCP}}}{n+1} > \frac{W_{k^{\text{WCP}}}}{W_{n+1}}$, meaning that $Q^{\text{CP}} < Q^{\text{WCP}}$. Here, we have $\frac{k^{\text{WCP}}}{n+1} > \frac{W_{k^{\text{WCP}}}}{W_{n+1}} \iff W_{n+1}k^{\text{WCP}} - (n+1)W_{k^{\text{WCP}}} > 0$.

$$(W_{k^{\text{WCP}}} + \Delta_{k^{\text{WCP}}})W_{n+1} \leq (W_{n+1} + \Delta_{n+1})W_{k^{\text{WCP}}}$$
$$W_{k^{\text{WCP}}}W_{n+1} + \Delta_{k^{\text{WCP}}}W_{n+1} \leq W_{n+1}W_{k^{\text{WCP}}} + \Delta_{n+1}W_{k^{\text{WCP}}}$$
$$\Delta_{k^{\text{WCP}}} \leq \frac{W_{k^{\text{WCP}}}}{W_{n+1}}\Delta_{n+1}$$
$$(\delta_{\max} - \delta_{\min})\tilde{\Delta}_{k^{\text{WCP}}} + k^{\text{WCP}}\delta_{\min} \leq \frac{W_{k^{\text{WCP}}}}{W_{n+1}}[(\delta_{\max} - \delta_{\min})\tilde{\Delta}_{n+1} + (n+1)\delta_{\min}]$$
$$\tilde{\Delta}_{k^{\text{WCP}}} + k^{\text{WCP}}\bar{\delta} \leq \frac{W_{k^{\text{WCP}}}}{W_{n+1}}[\tilde{\Delta}_{n+1} + (n+1)\bar{\delta}]$$
$$W_{n+1}\tilde{\Delta}_{k^{\text{WCP}}} + W_{n+1}k^{\text{WCP}}\bar{\delta} \leq W_{k^{\text{WCP}}}\tilde{\Delta}_{n+1} + W_{k^{\text{WCP}}}(n+1)\bar{\delta}$$
$$\bar{\delta}\left[W_{n+1}k^{\text{WCP}} - (n+1)W_{k^{\text{WCP}}}\right] \leq \tilde{\Delta}_{n+1}W_{k^{\text{WCP}}} - \tilde{\Delta}_{k^{\text{WCP}}}W_{n+1}$$
$$\bar{\delta} \leq \frac{\tilde{\Delta}_{n+1}W_{k^{\text{WCP}}} - \tilde{\Delta}_{k^{\text{WCP}}}W_{n+1}}{W_{n+1}k^{\text{WCP}} - (n+1)W_{k^{\text{WCP}}}}$$

We conclude the case $\frac{k^{\text{WCP}}}{n+1} > \frac{W_{k^{\text{WCP}}}}{W_{n+1}}$ by observing that

$$\hat{Q}^{\text{WCP}} \geq Q^{\text{WCP}} \iff \sum_{i=1}^{k^{\text{WCP}}} \hat{p}_i \leq \sum_{i=1}^{k^{\text{WCP}}} p_i \iff \frac{W_{k^{\text{WCP}}} + \Delta_{k^{\text{WCP}}}}{W_{n+1} + \Delta_{n+1}} \leq \frac{W_{k^{\text{WCP}}}}{W_{n+1}}$$

Notice that, according to the above derivations, the last inequality holds if and only if both requirement 1 and requirement 2 are satisfied, or both requirements are not satisfied. This is equivalent to requirement 1 NXOR requirement 2.

We now consider the case where $\frac{k^{\text{WCP}}}{n+1} < \frac{W_{k^{\text{WCP}}}}{W_{n+1}}$, that is that $Q^{\text{CP}} > Q^{\text{WCP}}$. Here, $\frac{k^{\text{WCP}}}{n+1} < \frac{W_{k^{\text{WCP}}}}{W_{n+1}} \iff W_{n+1}k^{\text{WCP}} - (n+1)W_{k^{\text{WCP}}} < 0$. This case is almost identical to the previous one, except for the following:

$$(W_{k^{\text{WCP}}} + \Delta_{k^{\text{WCP}}})W_{n+1} \leq (W_{n+1} + \Delta_{n+1})W_{k^{\text{WCP}}} \iff \bar{\delta} \geq \frac{\tilde{\Delta}_{n+1}W_{k^{\text{WCP}}} - \tilde{\Delta}_{k^{\text{WCP}}}W_{n+1}}{W_{n+1}k^{\text{WCP}} - (n+1)W_{k^{\text{WCP}}}}$$

Also, according to the same reasoning, we know that:

$$\hat{Q}^{\text{WCP}} \geq Q^{\text{WCP}} \iff \frac{W_{k^{\text{WCP}}} + \Delta_{k^{\text{WCP}}}}{W_{n+1} + \Delta_{n+1}} \leq \frac{W_{k^{\text{WCP}}}}{W_{n+1}}$$

Therefore, $\hat{Q}^{\text{WCP}} \geq Q^{\text{WCP}}$ holds if and only if exactly one of requirement 1 and requirement 2 is satisfied. This is equivalent to requirement 1 XOR requirement 2.

We now turn to analyze the setting where $\frac{k^{\text{WCP}}}{n+1} = \frac{W_{k^{\text{WCP}}}}{W_{n+1}}$.

$$(W_{k^{\text{WCP}}} + \Delta_{k^{\text{WCP}}})W_{n+1} \leq (W_{n+1} + \Delta_{n+1})W_{k^{\text{WCP}}} \iff$$
$$\bar{\delta}[W_{n+1}k^{\text{WCP}} - (n+1)W_{k^{\text{WCP}}}] \leq \tilde{\Delta}_{n+1}W_{k^{\text{WCP}}} - \tilde{\Delta}_{k^{\text{WCP}}}W_{n+1} \iff$$
$$0 \leq \tilde{\Delta}_{n+1}W_{k^{\text{WCP}}} - \tilde{\Delta}_{k^{\text{WCP}}}W_{n+1} \iff$$
$$\tilde{\Delta}_{k^{\text{WCP}}}W_{n+1} \leq \tilde{\Delta}_{n+1}W_{k^{\text{WCP}}} \iff$$
$$\frac{\tilde{\Delta}_{k^{\text{WCP}}}}{\tilde{\Delta}_{n+1}} \leq \frac{W_{k^{\text{WCP}}}}{W_{n+1}}$$

Notice that requirement 3 is defined as the last inequality. We therefore conclude that $\hat{Q}^{\text{WCP}} \geq Q^{\text{WCP}}$ holds if and only if requirement 1 NXOR requirement 3 is satisfied. $\qquad\square$

We now present a stochastic result that states the conditions under which WCP achieves a valid coverage rate when applied with weights that have a general error. The randomness is taken over random draws of $(X_{n+1}, Y_{n+1})$ from $P_{X,Y|Z}$.

**Proposition 3.** *Suppose that the calibration set is fixed, and equals to $\{(X_i, Y_i, Z_i) = (x_i, y_i, z_i)\}_{i=1}^n$ for some $x_i \in \mathcal{X}, y_i \in \mathcal{Y}, z_i \in \mathcal{Z}$ for all $i \in \{1, \ldots, n\}$. Further suppose that the test PI and the errors $\{\delta_i\}_{i=1}^{n+1}$ are also fixed, i.e., $Z_{n+1} = z$ for some $z \in \mathcal{Z}$. Then, $\hat{C}^{\text{WCP}}(X_{n+1})$, as defined in (11), achieves a conservative coverage rate, i.e,*

$$\mathbb{P}(Y_{n+1} \in \hat{C}^{\text{WCP}}(X_{n+1}) \mid Z_{n+1} = z, \{(X_i, Y_i, Z_i) = (x_i, y_i, z_i)\}_{i=1}^n)$$
$$\geq \mathbb{P}(Y_{n+1} \in C^{\text{WCP}}(X_{n+1}) \mid Z_{n+1} = z, \{(X_i, Y_i, Z_i) = (x_i, y_i, z_i)\}_{i=1}^n)$$

*if and only if one of the following is satisfied:*

1. *$k^{CP} < k^{\text{WCP}}$ and (requirement 1 NXOR requirement 2)*

2. *$k^{CP} > k^{\text{WCP}}$ and (requirement 1 XOR requirement 2)*

3. *$k^{CP} = k^{\text{WCP}}$ and (requirement 1 NXOR requirement 3)*

*Furthermore, if the above holds with probability at least $1 - \varepsilon$ over the random draws of $Z_{n+1}$, then we get a high marginal coverage rate:*

$$\mathbb{P}(Y_{n+1} \in \hat{C}^{\text{WCP}}(X_{n+1}) \mid \{(X_i, Y_i, Z_i) = (x_i, y_i, z_i)\}_{i=1}^n)$$
$$\geq \mathbb{P}(Y_{n+1} \in C^{\text{WCP}}(X_{n+1}) \mid \{(X_i, Y_i, Z_i) = (x_i, y_i, z_i)\}_{i=1}^n) - \varepsilon.$$

*Proof.* The proof is identical to the proof of Proposition 1 except for applying Lemma 2 instead of Lemma 1 and hence omitted. $\qquad\square$

We now present a different stochastic result in which the randomness is taken over random splits of the calibration and test samples.

**Proposition 4.** *Suppose that the calibration and test samples are given by $\{(X_i, Y_i, Z_i)\}_{i=1}^{n+1}$. Further suppose that:*

1. *$\mathbb{P}\left(k^{CP} < k^{\text{WCP}}, (\text{requirement 1 NXOR requirement 2})\right) \geq 1 - \varepsilon$,*

2. *$\mathbb{P}\left(k^{CP} > k^{\text{WCP}}, (\text{requirement 1 XOR requirement 2})\right) \geq 1 - \varepsilon$,*

3. *$\mathbb{P}\left(k^{CP} = k^{\text{WCP}}, (\text{requirement 1 NXOR requirement 3})\right) \geq 1 - \varepsilon$*

*Then, we get a high marginal coverage rate:*

$$\mathbb{P}(Y_{n+1} \in \hat{C}^{\text{WCP}}(X_{n+1})) \geq \mathbb{P}(Y_{n+1} \in C^{\text{WCP}}(X_{n+1})),$$

*Proof.* The proof is identical to the proof of Proposition 2 except for applying Lemma 2 instead of Lemma 1 and marginalizing over the randomness of the errors, and hence omitted. $\qquad\square$

The above theory sets the ground for Theorem 3.

*Proof of Theorem 3.* The proof is identical to the proof of Theorem 2 except for applying Proposition 4 instead of Proposition 2, and marginalizing over the randomness of the errors, and hence omitted. $\qquad\square$

## A.2 UNCERTAIN IMPUTATION

**Lemma 3.** *Denote the prediction interval function $C(x) = [a(x), b(x)]$. Suppose that $Y$ follows the model:*

$$Y = g^*(X, Z) + \varepsilon,$$

*where $\varepsilon$ is drawn from a distribution $P_{E^*}$ and $\varepsilon \perp\!\!\!\perp X \mid Z$. Suppose that*

1. *There exists a random variable $R^{test}$ drawn from a distribution $P_D$ such that:*

   (a) *$\hat{g}(X^{test}, Z^{test}) = g^*(X^{test}, Z^{test}) + R^{test}$,*
   (b) *$R^{test} \perp\!\!\!\perp g^*(X^{test}, Z^{test}) \mid Z^{test}$,*
   (c) *$R^{test} \perp\!\!\!\perp C(X^{test}) \mid Z^{test}$.*

2. *For every $z \in \mathcal{Z}$ and $x \in \mathcal{X}$ such that $f_{X^{test}, Z^{test}}(x, z) > 0$ the density of $Y^{test} \mid X^{test} = x, Z^{test} = z$ is peaked inside the interval $C(x) = [a(x), b(x)]$, i.e.,*

   (a) *$\forall v > 0 : f_{Y^{test} \mid X^{test}=x, Z^{test}=z}(b(x) + v) \leq f_{Y^{test} \mid X^{test}=x, Z^{test}=z}(b(x) - v)$, and*
   (b) *$\forall v > 0 : f_{Y^{test} \mid X^{test}=x, Z^{test}=z}(a(x) - v) \leq f_{Y^{test} \mid X^{test}=x, Z^{test}=z}(a(x) + v)$.*

*Suppose that the test PI is fixed and equals to $Z^{test} = z$ for some $z \in \mathcal{Z}$. Denote the imputed test variable:*

$$\bar{Y}^{test} = \hat{g}(X^{test}, Z^{test}) + e$$

*where $e$ is a random variable drawn from the same distribution as $Y^{test} - \hat{g}(X^{test}, Z^{test}) \mid Z^{test} = z$. Then,*

$$\mathbb{P}(Y^{test} \in C(X^{test}) \mid Z^{test} = z) \geq \mathbb{P}(\bar{Y}^{test} \in C(X^{test}) \mid Z^{test} = z).$$

*Above, the probability is taken over draws of the test variables: $(X^{test}, Z^{test}, Y^{test}, \varepsilon^{test}) \sim P_{X,Z,Y,E^*}$. If we further assume that $R^{test} \perp\!\!\!\perp X^{test}$, i.e., $R^{test}$ is a deterministic function of $Z^{test}$, then we obtain coverage equality:*

$$\mathbb{P}(Y^{test} \in C(X^{test}) \mid Z^{test} = z) = \mathbb{P}(\bar{Y}^{test} \in C(X^{test}) \mid Z^{test} = z).$$

*Proof.* All formulations in this proof are conducted conditioned on $Z^{\text{test}} = z$. For ease of notation, we omit this conditioning, yet emphasize that the probabilities are taken over draws conditional on $Z^{\text{test}} = z$. From the definition of $e$, and under the model of $Y$, there exist $(X, Y, D', \varepsilon) \sim P_{X,Y,D,E^*}$ such that:

$$e = Y - \hat{g}(X, z) = g^*(X, z) + \varepsilon - \hat{g}(X, z) = g^*(X, z) + \varepsilon - (g^*(X, z) + D') = \varepsilon - D'.$$

We remark that $\varepsilon \perp\!\!\!\perp X^{\text{test}}$, $D' \perp\!\!\!\perp X^{\text{test}}$ and $\varepsilon \perp\!\!\!\perp Y^{\text{test}}$ since these are drawn independently of $X^{\text{test}}$. Following the assumption on $Y$, there exists $\varepsilon^{\text{test}}$ which satisfies:

$$Y^{\text{test}} = g^*(X^{\text{test}}, Z^{\text{test}}) + \varepsilon^{\text{test}},$$

where $\varepsilon^{\text{test}} \sim P_{E^*}$ and $\varepsilon^{\text{test}} \perp\!\!\!\perp X^{\text{test}}$. Therefore, the variable $Y'$, formulated as:

$$Y' := g^*(X^{\text{test}}, Z^{\text{test}}) + \varepsilon$$

is equal in distribution to $Y^{\text{test}}$, that is: $Y' \stackrel{d}{=} Y^{\text{test}}$. Furthermore, since $\varepsilon^{\text{test}} \perp\!\!\!\perp X^{\text{test}}$ we also get that:

$$Y' \mid X^{\text{test}} \stackrel{d}{=} Y^{\text{test}} \mid X^{\text{test}}.$$

We denote by $R$ the sum of $R^{\text{test}}$ and $-D'$:

$$R := R^{\text{test}} - D'.$$

Observe that since $R^{\text{test}} \stackrel{d}{=} D'$ we get that $R$ is a symmetric random variable:

$$\mathbb{P}(R \leq r) = \mathbb{P}(R^{\text{test}} - D' \leq r) = \mathbb{P}(D' - R^{\text{test}} \leq r) = \mathbb{P}(-R \leq r).$$

Moreover, the mean of $R$ is 0:

$$\mathbb{E}[R] = \mathbb{E}[R^{\text{test}} - D'] = \mathbb{E}[R^{\text{test}}] - \mathbb{E}[D'] = 0.$$

Since $R$ is a symmetric random variable with mean 0, its median is 0 as well. We now develop the imputed value $\bar{Y}^{\text{test}}$:

$$
\begin{aligned}
\bar{Y}^{\text{test}} &= \hat{g}(X^{\text{test}}, Z^{\text{test}}) + e \\
&= \hat{g}(X^{\text{test}}, Z^{\text{test}}) + \varepsilon - D' \\
&= g^*(X^{\text{test}}, Z^{\text{test}}) + R^{\text{test}} + \varepsilon - D' \\
&= Y' + R.
\end{aligned}
$$

Suppose that $C(X^{\text{test}}) = [a(X^{\text{test}}), b(X^{\text{test}})]$ is a prediction interval for $Y^{\text{test}}$. We now compute the probability $\mathbb{P}(\bar{Y}^{\text{test}} \leq b(X^{\text{test}}))$. For ease of notation, we denote $b(X^{\text{test}})$ by $B$.

$$
\begin{aligned}
\mathbb{P}(\bar{Y}^{\text{test}} \leq B) &= \mathbb{P}(Y' + R \leq B) \\
&= \mathbb{P}(g^*(X^{\text{test}}, Z^{\text{test}}) + \varepsilon + R \leq B) \\
&= \mathbb{P}(g^*(X^{\text{test}}, Z^{\text{test}}) + \varepsilon^{\text{test}} + R \leq B) \\
&= \mathbb{P}(Y^{\text{test}} + R \leq B) \\
&= \mathbb{P}(Y^{\text{test}} + R \leq B \mid R \geq 0)\mathbb{P}(R \geq 0) + \mathbb{P}(Y^{\text{test}} + R \leq B \mid R < 0)\mathbb{P}(R < 0) \\
&= \frac{1}{2}\left[\mathbb{P}(Y^{\text{test}} + R \leq B \mid R \geq 0) + \mathbb{P}(Y^{\text{test}} + R \leq B \mid R < 0)\right] \\
&= \frac{1}{2}\left[\mathbb{P}(Y^{\text{test}} + R \leq B \mid R \geq 0) + \mathbb{P}(Y^{\text{test}} - R \leq B \mid R \geq 0)\right] \\
&= \frac{1}{2}\left[\mathbb{P}(Y^{\text{test}} \leq B - R \mid R \geq 0) + \mathbb{P}(Y^{\text{test}} \leq B + R \mid R \geq 0)\right] \\
&= \frac{1}{2}\int_{r \geq 0}\left[\mathbb{P}(Y^{\text{test}} \leq B - R \mid R = r) + \mathbb{P}(Y^{\text{test}} \leq B + R \mid R = r)\right] f_{R \mid R \geq 0}(r)dr.
\end{aligned}
$$

We now turn to develop $\mathbb{P}(Y^{\text{test}} \leq B - R \mid R = r) + \mathbb{P}(Y^{\text{test}} \leq B + R \mid R = r)$. We begin by conditioning on $X^{\text{test}}$ being equal to some $x \in \mathcal{X}$. Therefore, we can write:

$$
\begin{aligned}
\mathbb{P}(Y^{\text{test}} &\leq B - R \mid R = r, X^{\text{test}} = x) + \mathbb{P}(Y^{\text{test}} \leq B + R \mid R = r, X^{\text{test}} = x) \\
&= 2\mathbb{P}(Y^{\text{test}} \leq B \mid R = r, X^{\text{test}} = x) \\
&\quad + \mathbb{P}(Y^{\text{test}} \leq B - R \mid R = r, X^{\text{test}} = x) - \mathbb{P}(Y^{\text{test}} \leq B \mid R = r, X^{\text{test}} = x) \\
&\quad + \mathbb{P}(Y^{\text{test}} \leq B + R \mid R = r, X^{\text{test}} = x) - \mathbb{P}(Y^{\text{test}} \leq B \mid R = r, X^{\text{test}} = x) \\
&= 2\mathbb{P}(Y^{\text{test}} \leq B \mid R = r, X^{\text{test}} = x) \\
&\quad + \mathbb{P}(B \leq Y^{\text{test}} \leq B + R \mid R = r, X^{\text{test}} = x) \\
&\quad - \mathbb{P}(B - R \leq Y^{\text{test}} \leq B \mid R = r, X^{\text{test}} = x).
\end{aligned}
$$

Observe that $Y^{\text{test}} \perp\!\!\!\perp R$ since $g^*(X^{\text{test}}, z)$ is independent of $R^{\text{test}}$. Therefore, we can omit the conditioning on $R = r$:

$$
\begin{aligned}
\mathbb{P}(B \leq Y^{\text{test}} &\leq B + R \mid R = r, X^{\text{test}} = x) - \mathbb{P}(B - R \leq Y^{\text{test}} \leq B \mid R = r, X^{\text{test}} = x) \\
&= \mathbb{P}(B \leq Y^{\text{test}} \leq B + r \mid X^{\text{test}} = x) - \mathbb{P}(B - r \leq Y^{\text{test}} \leq B \mid X^{\text{test}} = x).
\end{aligned}
$$

Since the density of $Y^{\text{test}} \mid X^{\text{test}}$ is assumed to peak inside the interval, we get:

$$
\mathbb{P}(B \leq Y^{\text{test}} \leq B + r \mid X^{\text{test}} = x) - \mathbb{P}(B - r \leq Y^{\text{test}} \leq B \mid X^{\text{test}} = x) \leq 0.
$$

Therefore:

$$
\begin{aligned}
\mathbb{P}(Y^{\text{test}} \leq B - R \mid R = r, X^{\text{test}} = x) &+ \mathbb{P}(Y^{\text{test}} \leq B + R \mid R = r, X^{\text{test}} = x) \\
&\leq 2\mathbb{P}(Y^{\text{test}} \leq B \mid R = r, X^{\text{test}} = x).
\end{aligned}
$$

We return to the marginal statement:

$$\mathbb{P}(Y^{\text{test}} \leq B - R \mid R = r) + \mathbb{P}(Y^{\text{test}} \leq B + R \mid R = r)$$
$$= \int_{x \in \mathcal{X}} [\mathbb{P}(Y^{\text{test}} \leq B - R \mid R = r, X^{\text{test}} = x)$$
$$+ \mathbb{P}(Y^{\text{test}} \leq B + R \mid R = r, X^{\text{test}} = x)] f_{X^{\text{test}} \mid R = r}(x; r) dx$$
$$\leq \int_{x \in \mathcal{X}} [2\mathbb{P}(Y^{\text{test}} \leq B \mid R = r, X^{\text{test}} = x)] f_{X^{\text{test}} \mid R = r}(x; r) dx$$
$$= 2\mathbb{P}(Y^{\text{test}} \leq B \mid R = r).$$

We plug this in and get:

$$\mathbb{P}(\bar{Y}^{\text{test}} \leq B) = \frac{1}{2} \int_{r \geq 0} \left[ \mathbb{P}(Y^{\text{test}} \leq B - R \mid R = r) \right.$$
$$\left. + \mathbb{P}(Y^{\text{test}} \leq B + R \mid R = r) \right] f_{R \mid R \geq 0}(r) dr$$
$$\leq \frac{1}{2} \int_{r \geq 0} \left[ 2\mathbb{P}(Y^{\text{test}} \leq B \mid R = r) \right] f_{R \mid R \geq 0}(r) dr$$
$$= \mathbb{P}(Y^{\text{test}} \leq B \mid R \geq 0)$$
$$= \mathbb{P}(Y^{\text{test}} \leq B).$$

The last equality holds since $Y^{\text{test}} \perp\!\!\!\perp (R^{\text{test}}, D') \mid Z^{\text{test}}$ and $C(X^{\text{test}}) \perp\!\!\!\perp (R^{\text{test}}, D') \mid Z^{\text{test}}$. The proof for $\mathbb{P}(\bar{Y}^{\text{test}} \geq A) \leq \mathbb{P}(Y^{\text{test}} \geq A)$ is similar and hence omitted. By combining these we get:

$$\mathbb{P}(\bar{Y}^{\text{test}} \in C(X^{\text{test}})) = \mathbb{P}(a(X^{\text{test}}) \leq \bar{Y}^{\text{test}} \leq b(X^{\text{test}}))$$
$$= \mathbb{P}(\bar{Y}^{\text{test}} \leq b(X^{\text{test}})) - \mathbb{P}(\bar{Y}^{\text{test}} \geq a(X^{\text{test}}))$$
$$\leq \mathbb{P}(Y^{\text{test}} \leq b(X^{\text{test}})) - \mathbb{P}(Y^{\text{test}} \geq a(X^{\text{test}}))$$
$$= \mathbb{P}(a(X^{\text{test}}) \leq Y^{\text{test}} \leq b(X^{\text{test}}))$$
$$= \mathbb{P}(Y^{\text{test}} \in C(X^{\text{test}})).$$

We now consider the setting where $R^{\text{test}} \perp\!\!\!\perp X^{\text{test}}$, i.e., $R^{\text{test}}$ is a deterministic function of $Z^{\text{test}}$. Since we conditioned on $Z^{\text{test}} = z$ in this analysis, we get that $R^{\text{test}} = D'$ are constants. Therefore:

$$\bar{Y} = Y' + R = Y' + R^{\text{test}} - D' = Y'.$$

Which leads to:

$$\mathbb{P}(\bar{Y}^{\text{test}} \in C(X^{\text{test}})) = \mathbb{P}(Y' \in C(X^{\text{test}})) = \mathbb{P}(Y^{\text{test}} \in C(X^{\text{test}}))$$

$\square$

**Lemma 4.** *Suppose that $C(x)$ is a prediction set satisfying the assumptions of Lemma 3. Denote the imputed variable by:*

$$\bar{Y}^{test} := \begin{cases} Y^{test}, & M^{test} = 0, \\ \hat{g}(X^{test}, Z^{test}) + e, & M^{test} = 1, \end{cases}$$

*where $e$ is a random variable drawn from the distribution of $Y^{test} - \hat{g}(X^{test}, Z^{test}) \mid Z^{test}$. Then, under the assumptions of Lemma 3, and assuming $(X^{test}, Y^{test}) \perp\!\!\!\perp M^{test} \mid Z^{test}$, we get:*

$$\mathbb{P}(Y^{test} \in C(X^{test})) \geq \mathbb{P}(\bar{Y}^{test} \in C(X^{test})).$$

*If we further assume that $R^{test} \perp\!\!\!\perp X^{test}$, i.e., $R^{test}$ is a deterministic function of $Z^{test}$, then we obtain coverage equality:*

$$\mathbb{P}(Y^{test} \in C(X^{test})) = \mathbb{P}(\bar{Y}^{test} \in C(X^{test})).$$

*Proof.* All formulations in this proof are conducted conditioned on $Z^{\text{test}} = z$ unless explicitly stated otherwise. For ease of notation, we omit this conditioning, while the probabilities are taken of draws

conditional on $Z^{\text{test}} = z$. By applying Lemma 3 and assuming $(X^{\text{test}}, Y^{\text{test}}) \perp\!\!\!\perp M^{\text{test}} \mid Z^{\text{test}}$ we get:

$$
\begin{aligned}
\mathbb{P}(\bar{Y}^{\text{test}} \in C(X^{\text{test}})) &= \mathbb{P}(\bar{Y}^{\text{test}} \in C(X^{\text{test}}) \mid M^{\text{test}} = 0)\mathbb{P}(M^{\text{test}} = 0) \\
&\quad + \mathbb{P}(\bar{Y}^{\text{test}} \in C(X^{\text{test}}) \mid M^{\text{test}} = 1)\mathbb{P}(M^{\text{test}} = 1) \\
&= \mathbb{P}(Y^{\text{test}} \in C(X^{\text{test}}) \mid M^{\text{test}} = 0)\mathbb{P}(M^{\text{test}} = 0) \\
&\quad + \mathbb{P}(\hat{g}(X^{\text{test}}, Z^{\text{test}}) + e \in C(X^{\text{test}}) \mid M^{\text{test}} = 1)\mathbb{P}(M^{\text{test}} = 1) \\
&= \mathbb{P}(Y^{\text{test}} \in C(X^{\text{test}}))\mathbb{P}(M^{\text{test}} = 0) \\
&\quad + \mathbb{P}(\hat{g}(X^{\text{test}}, Z^{\text{test}}) + e \in C(X^{\text{test}}))\mathbb{P}(M^{\text{test}} = 1) \\
&\leq \mathbb{P}(Y^{\text{test}} \in C(X^{\text{test}}))\mathbb{P}(M^{\text{test}} = 0) \\
&\quad + \mathbb{P}(Y^{\text{test}} \in C(X^{\text{test}}))\mathbb{P}(M^{\text{test}} = 1) \\
&= \mathbb{P}(Y^{\text{test}} \in C(X^{\text{test}}) \mid M^{\text{test}} = 0)\mathbb{P}(M^{\text{test}} = 0) \\
&\quad + \mathbb{P}(Y^{\text{test}} \in C(X^{\text{test}}) \mid M^{\text{test}} = 1)\mathbb{P}(M^{\text{test}} = 1) \\
&= \mathbb{P}(Y^{\text{test}} \in C(X^{\text{test}}))
\end{aligned}
$$

We now return to explicitly stating the conditioning on $Z^{\text{test}}$. Thus far, we showed that for every $z \in \mathcal{Z}$:

$$
\mathbb{P}(\bar{Y}^{\text{test}} \in C(X^{\text{test}}) \mid Z^{\text{test}} = z) \leq \mathbb{P}(Y^{\text{test}} \in C(X^{\text{test}}) \mid Z^{\text{test}} = z).
$$

By marginalizing this, we obtain:

$$
\mathbb{P}(\bar{Y}^{\text{test}} \in C(X^{\text{test}})) \leq \mathbb{P}(Y^{\text{test}} \in C(X^{\text{test}})).
$$

When assuming that $R^{\text{test}} \perp\!\!\!\perp X^{\text{test}}$, by Lemma 3 we get

$$
\mathbb{P}(\hat{g}(X^{\text{test}}, Z^{\text{test}}) + e \in C(X^{\text{test}}) \mid Z^{\text{test}} = z) = \mathbb{P}(\bar{Y}^{\text{test}} \in C(X^{\text{test}}) \mid Z^{\text{test}} = z).
$$

By plugging in this expression to the development of $\mathbb{P}(\bar{Y}^{\text{test}} \in C(X^{\text{test}}))$ above, we get:

$$
\mathbb{P}(\bar{Y}^{\text{test}} \in C(X^{\text{test}}) \mid Z^{\text{test}} = z) = \mathbb{P}(\bar{Y}^{\text{test}} \in C(X^{\text{test}}) \mid Z^{\text{test}} = z).
$$

By taking the expectation over $Z^{\text{test}}$ we obtain:

$$
\mathbb{P}(\bar{Y}^{\text{test}} \in C(X^{\text{test}})) = \mathbb{P}(\bar{Y}^{\text{test}} \in C(X^{\text{test}})).
$$

$\square$

Armed with the above theory, we now prove Theorem 4

*Proof of Theorem 4.* Denote the imputed variable by:

$$
\bar{Y}^{\text{test}} := \begin{cases} Y^{\text{test}}, & M^{\text{test}} = 0, \\ \hat{g}(X^{\text{test}}, Z^{\text{test}}) + e, & M^{\text{test}} = 1. \end{cases}
$$

Above, $e$ is a random variable drawn from the distribution of $Y - \hat{g}(X, Z) \mid Z = Z^{\text{test}}, M = 0$. Since $(X, Y) \perp\!\!\!\perp M \mid Z$, this distribution is equivalent to the distribution of $Y - \hat{g}(X, Z) \mid Z = Z^{\text{test}}$. Since $C^{\text{UI}}$ is constructed by CP using the imputed labels, and due to the exchangeability of the data, $C^{\text{UI}}$ covers $\bar{Y}^{\text{test}}$ at the desired coverage rate Vovk et al. (2005); Angelopoulos & Bates (2023):

$$
\mathbb{P}(\bar{Y}^{\text{test}} \in C^{\text{UI}}(X^{\text{test}})) \geq 1 - \alpha.
$$

By applying Lemma 4, and since the samples are drawn independently, we get:

$$
\mathbb{P}(Y^{\text{test}} \in C^{\text{UI}}(X^{\text{test}}) \mid \{(X_i, Y_i, \tilde{Y}_i, Z_i, M_i)\}_{i=1}^n) \geq \mathbb{P}(\bar{Y}^{\text{test}} \in C^{\text{UI}}(X^{\text{test}}) \mid \{(X_i, Y_i, \tilde{Y}_i, Z_i, M_i)\}_{i=1}^n).
$$

By marginalizing the above, we obtain:

$$
\mathbb{P}(Y^{\text{test}} \in C^{\text{UI}}(X^{\text{test}})) \geq \mathbb{P}(\bar{Y}^{\text{test}} \in C^{\text{UI}}(X^{\text{test}})) \geq 1 - \alpha.
$$

$\square$

### A.3 TRIPLY ROBUST CALIBRATION

Building on the validity of `UI`, we turn to prove the validity of the triply robust method.

*Proof of Theorem 5.* Under the assumptions of Theorem 1 we get $\mathbb{P}(Y^{\text{test}} \in C^{\text{PCP}}(X^{\text{test}})) \geq 1 - \alpha$, under the assumptions of Theorem 4 we get $\mathbb{P}(Y^{\text{test}} \in C^{\text{PCP}}(X^{\text{test}})) \geq 1 - \alpha$. By Vovk et al. (2005); Angelopoulos & Bates (2023), CP constructs prediction sets with the nominal coverage level when the scores are exchangeable, i.e., $\mathbb{P}(Y^{\text{test}} \in C^{\text{CP}}(X^{\text{test}})) \geq 1 - \alpha$. Thus, when at least one of the above assumptions hold, one of the above prediction sets achieves the desired coverage rate. Since `TriplyRobust` is the union of all three sets, its coverage rate is greater than or equal to the coverage rate of each prediction set. Therefore:

$$\mathbb{P}(Y^{\text{test}} \in C^{\texttt{TriplyRobust}}(X^{\text{test}})) \geq 1 - \alpha$$

$\square$

## B ADDITIONAL RELATED WORK

Practical examples for tasks with privileged information include crowd-sourcing settings, such as CIFAR-10H, in which annotator metadata, e.g., response times, confidence scores, and expertise level, may serve as PI. Another example is e-commerce recommendation, where a user's click history can be considered as PI, which is a good predictor for actual purchases (Yang et al., 2022). In this example, vendors may not share the user's click history with the model provider due to privacy issues, which makes the PI unavailable at test time. In medical imaging, a pathologist's diagnostic report for a biopsy image can act as privileged information that strongly predicts whether the tissue is cancerous or healthy (Lopez-Paz et al., 2015). In this context, the PI may not be available for all patients at test time for various reasons, such as limited resources or prioritization.

Several recent works extend conformal prediction beyond the i.i.d. assumption to handle various forms of distribution shifts, including arbitrary shifts (Barber et al., 2023), online and time-series settings (Gibbs & Candes, 2021; Gibbs & Candès, 2024; Zaffran et al., 2022; Feldman et al., 2023; Xu & Xie, 2023), missing covariates (Zaffran et al., 2023; 2024), and ambiguous labels (Stutz et al., 2023; Caprio et al., 2024). Related research on missing-data imputation includes analyses of missingness mechanisms (Rubin, 1976), and multiple imputation frameworks (Rubin, 1996; 2018). Furthermore, the work of Zhang (2016) studies regression imputation approaches and discusses adding residual variance to account for prediction uncertainty.

## C ALGORITHMS

### C.1 PRIVILEGED CONFORMAL PREDICTION

Algorithm 1 given below details the `PCP` method.

### C.2 NAIVE IMPUTATION

A natural approach to handle corrupted labels is to impute them, using the observed covariates and privileged information. Then, CP can be simply employed using the imputed labels. Formally, we begin similarly to CP and PCP and split the data into two parts: a training set, $\mathcal{I}_1$, and a calibration set, $\mathcal{I}_2$. Then, we fit two predictive models to estimate the response using the training data: a model $\hat{f}(x)$ which takes as an input the feature vector $X$, and a label imputator $\hat{g}(x, z)$ which takes as an input the feature vector $X$ and the privileged information $Z$. Next, we impute the corrupted labels in the calibration set using the model $\hat{g}$:

$$\bar{Y}_i := \begin{cases} Y_i & \text{if } M_i = 0, \\ \hat{g}(X_i, Z_i) & \text{otherwise} \end{cases}, \forall i \in \mathcal{I}_2$$

We compute the non-conformity scores using the imputed labels: $\bar{S}_i = \mathcal{S}(X_i, \bar{Y}_i; \hat{f}), \forall i \in \mathcal{I}_2$. The scores threshold is defined using the above scores:

$$Q^{\texttt{NaiveImpute}} := \text{Quantile}\left(1 - \alpha; \sum_{i \in \mathcal{I}_2} \frac{1}{|\mathcal{I}_2| + 1}\delta_{\bar{S}_i} + \frac{1}{|\mathcal{I}_2| + 1}\delta_\infty\right).$$

---

**Algorithm 1:** Privileged Conformal Prediction (PCP)

---

**Input:**

Data $(X_i, \tilde{Y}_i, Z_i, M_i) \in \mathcal{X} \times \mathcal{Y} \times \mathcal{Z} \times \{0, 1\}, 1 \leq i \leq n$, weights $\{w_i\}_{i=1}^n$, miscoverage level $\alpha \in (0, 1)$, level $\beta \in (0, \alpha)$, an algorithm $\hat{f}(x)$, a score function $\mathcal{S}$, and a test point $X^{\text{test}} = x$.

**Process:**

Randomly split $\{1, \ldots, n\}$ into two disjoint sets $\mathcal{I}_1, \mathcal{I}_2$.

Fit the base algorithm $\hat{f}$ on the training data $\{(X_i, \tilde{Y}_i)\}_{i \in \mathcal{I}_1}$.

Compute the scores $S_i = \mathcal{S}(X_i, \tilde{Y}_i; \hat{f})$ for the calibration samples, $i \in \mathcal{I}_2$.

Compute the normalized weights:

$$p_j^i = \frac{w_j}{\sum_{k \in \mathcal{I}_2^{\text{uc}}} w_k + w_i}$$

Compute a threshold $Q(Z_i)$ for each calibration sample:

$$Q(Z_i) = \text{Quantile}\left(1 - \alpha + \beta; \sum_{j \in \mathcal{I}_2^{\text{uc}}} p_j^i \delta_{S_j} + p_i^i \delta_\infty\right)$$

Compute $Q^{\text{PCP}}$, the $(1 - \beta)$ quantile of $\{Q(Z_i)\}_{i \in \mathcal{I}_2}$:

$$Q^{\text{PCP}} = \text{Quantile}\left(1 - \beta; \sum_{j \in \mathcal{I}_2} \frac{1}{|\mathcal{I}_2| + 1} \delta_{Q(Z_i)} + \frac{1}{|\mathcal{I}_2| + 1} \delta_\infty\right)$$

**Output:**

Prediction set $C^{\text{PCP}}(x) = \{y : \mathcal{S}(x, y; \hat{f}) \leq Q^{\text{PCP}}\}$.

---

Finally, for a new input data $X^{\text{test}}$, we construct the prediction set for $Y^{\text{test}}$ as follows:

$$C^{\text{NaiveImpute}}(X^{\text{test}}) = \left\{y : \mathcal{S}(X^{\text{test}}, y, \hat{f}) \leq Q^{\text{NaiveImpute}}\right\}.$$

While this approach is simple and intuitive, it does not hold any theoretical guarantees. Furthermore, the experiments in Section 4.3 reveal that it consistently undercovers the response. We believe that this is attributed to the fact that the imputed labels are estimates of $\mathbb{E}[Y \mid X, Z]$, which reduces the uncertainty of the imputed label. This, in turn, leads to narrower intervals.

### C.3 UNCERTAIN IMPUTATION

The UI algorithm is fully described in Algorithm C.3 below.

#### C.3.1 ERROR SAMPLING TECHNIQUES

In this section, we present our suggestions for sampling errors conditional on the privileged information for the imputation process of UI in (7). Instead of using a strict equality condition of $Z_i = z$ in $\mathcal{E}(z) := \{E_i : i \in \mathcal{I}_3, Z_i = z\}$, we can relax this requirement by clustering the $Z$ space and comparing the clusters into which each $z$ falls, that is, $\mathcal{E}(z) := \{E_i : i \in \mathcal{I}_3, h(Z_i) = h(z)\}$, where $h$ is a clustering function. The first clustering method we use is Kmeans, which we fit on the training and validation data. Then, we cluster the PIs of the reference set. When imputing the corrupted labels of the calibration set, we sample an error from the cluster corresponding to the test PI. The second clustering approach we examine is Linear clustering. We first fit on the training and validation data a linear model that takes as an input the PI $Z$ and outputs an estimated label $Y$. We used this model to compute the estimated labels for each point in the reference set and then split them into bins. We note that the alternative approach of learning the error distribution given the PI $Z$ may be applied as well, such as random forest for conditional density estimation (Pospisil & Lee, 2018), or normalizing flows (Rezende & Mohamed, 2015; Papamakarios et al., 2021).

---

**Algorithm 2:** Uncertain imputation (`UI`)

---

**Input:**

Data $(X_i, \tilde{Y}_i, Z_i, M_i) \in \mathcal{X} \times \mathcal{Y} \times \mathcal{Z} \times \{0, 1\}, 1 \leq i \leq n$, miscoverage level $\alpha \in (0, 1)$, an algorithm $\hat{f}(x)$, and algorithm $\hat{g}(x, z)$, a score function $\mathcal{S}$, and a test point $X^{\text{test}} = x$.

**Process:**

Randomly split $\{1, \ldots, n\}$ into three disjoint sets $\mathcal{I}_1, \mathcal{I}_2, \mathcal{I}_3$.

Fit the base algorithm $\hat{f}$ on the training data $\{(X_i, \tilde{Y}_i)\}_{i \in \mathcal{I}_1}$.

Fit the predictor $\hat{g}$ on the training data $\{(X_i, Z_i, \tilde{Y}_i)\}_{i \in \mathcal{I}_1}$.

Compute the errors $E_i$ of $\hat{g}$ on the reference set $\mathcal{I}_3$ according to (6).

Generate the imputed labels $\bar{Y}_i, \forall i \in \mathcal{I}_2$, according to (7).

Compute the scores $\bar{S}_i = \mathcal{S}(X_i, \bar{Y}_i; \hat{f})$ using the imputed labels for the calibration samples, $i \in \mathcal{I}_2$.

Compute $Q^{\text{UI}}$, the $(1 - \alpha)$ quantile of the scores

$$Q^{\text{UI}} = \text{Quantile}\left(1 - \alpha; \sum_{j \in \mathcal{I}_2} \frac{1}{|\mathcal{I}_2| + 1} \delta_{\bar{S}_i} + \frac{1}{|\mathcal{I}_2| + 1} \delta_\infty\right)$$

**Output:**

Prediction set $C^{\text{UI}}(x) = \{y : \mathcal{S}(x, y; \hat{f}) \leq Q^{\text{UI}}\}$.

---

## C.4 METHOD VALIDITY CONDITIONS SUMMARY

We summarize the validity conditions of each method in Table 1. This table illustrates that `TriplyRobust` achieves the desired coverage rate when at least one of the underlying methods is valid.

Table 1: Summary of Method Validity Conditions

| Method | Guarantee | Accurate $Y \mid X$ est. | Accurate $M \mid Z$ est. | Accurate $Y \mid Z$ est. |
|---|---|---|---|---|
| Quantile Regression | (Koenker & Bassett, 1978; Koenker, 2005; Steinwart & Christmann, 2011; Takeuchi et al., 2006) | ✓ | NA | NA |
| PCP | Theorem 1 | x | ✓ | NA |
| UI | Theorem 4 | x | NA | ✓ |
| TriplyRobust | Theorem 5 | ✓ | ✓ | ✓ |

# D DATASETS DETAILS

## D.1 GENERAL REAL DATASET DETAILS

Table 2 displays the size of each dataset, the feature dimension, and the feature that is used as privileged information in the tabular data experiments.

Table 2: Information about the real data sets.

| Dataset | # Samples | $X/Z/Y$ Dimensions | $Z$ description |
|---|---|---|---|
| facebook1 (facebook) | 40948 | 52/1/1 | Number of posts comments |
| facebook2 (facebook) | 81311 | 52/1/1 | Number of posts comments |
| Bio (bio) | 45730 | 8/1/1 | Fractional area of exposed non polar residue |
| House (house) | 21613 | 17/1/1 | Square footage of the apartments interior living space |
| Meps19 (meps_19) | 15785 | 138 /3/1 | Overall rating of feelings, age, working limitation |
| NSLM (Yeager et al., 2019) | 10391 | 10/1/1 | Synthetic normally distributed random variable |

### D.2 NSLM DATASET DETAILS

The 2018 Atlantic Causal Inference Conference workshop on heterogeneous treatment effects (Carvalho et al., 2019) introduced the National Study of Learning Mindsets (NSLM) dataset (Yeager et al., 2019). We refer to Carvalho et al. (2019, Section 2) for its full details. In our experiments, we adopt the approach outlined in Carvalho et al. (2019); Lei & Candès (2021) to generate synthetic potential outcomes and a synthetic PI variable. The process begins by standardizing the dataset so that all features have a mean of 0 and a standard deviation of 1. The data is then randomly split into two subsets: 80% samples are used for training, while the remaining 20% samples are used for validation. To model the relationship between $X$ and $Y$, we train a neural network with a single hidden layer containing 32 neurons. The learning mechanism of the network is described in Section E.1. We refer to its learned function as $\hat{\mu}_0(\cdot)$. Additionally, we employ an XGBoost classifier to estimate the original treatment variable $M$ using the features in $X$. The classifier is configured with a maximum depth of 2 and uses 10 estimators. We then calibrate the estimated propensity score to have the same marginal probability as the original probability and denote the calibrated score $\hat{e}(X_i)$. Once these models are trained, we generate a new treatment indicator $M_i$, a synthetic PI variable $Z_i$, and a semi-synthetic outcome variable $Y_i$ as detailed next:

$$Z_i \sim \mathcal{N}(0, 0.2^2)$$
$$E_i = \mathbb{1}\{Z_i \geq \text{Quantile}(0.9, Z) \text{ or } Z_i \leq \text{Quantile}(0.1, Z)\}$$
$$M_i \sim Ber(\min(0.8, (1 + E_i)\hat{e}(X_i)))$$
$$\tau_i = 0.228 + 0.05\mathbb{1}\{X_{i,5} < 0.07) - 0.05\mathbb{1}\{X_{i,6} < -0.69\} - 0.08\mathbb{1}\{X_{i,1} \in \{1, 13, 14\}\}$$
$$Y_i = \hat{\mu}_0(X_i) + \tau_i + (1 + E_i)Z_i.$$

### D.3 SYNTHETIC DATASET DETAILS

In this section, we present the synthetic datasets used in this work. Across all datasets, $X, Y, Z$ are generated using the same procedure from Feldman & Romano (2024), where the only difference is the label corruption mechanism. We first describe the generation process for $X, Y, Z$ and then detail the corruption mechanisms.

The feature vectors are uniformly sampled as follows:

$$X_i \sim \text{Uni}(1, 5)^{10},$$

where $\text{Uni}(a, b)$ is a unifrom distribution in the range $(a, b)$. The $p = 3$ dimensional PI $Z_{i,j}$, for each dimension $j \in \{1, \ldots, 3\}$ is sampled as:

$$E_{i,j}^1 \sim \mathcal{N}(0, 1),$$
$$E_{i,j}^2 \sim \text{Uni}(-1, 1),$$
$$E_{i,j}^3 \sim \mathcal{N}(0, 1),$$
$$P_{i,j} \sim \text{Pois}(\cos(E_i^1)^2 + 0.1) * E_i^2,$$
$$Z_{i,j} \sim P_i + 2E_i^3.$$

Above, $\text{Pois}(\lambda)$ is a poisson distribution with parameter $\lambda$, and $\mathcal{N}(\mu, \sigma^2)$ is a normal distribution with mean $\mu$ and variance $\sigma^2$. We define some additional variables:

$$\beta_1 \sim \text{Uni}(0,1)^1$$
$$\beta_1 = \beta_1 / ||\beta_1||_1$$
$$\beta_2 \sim \text{Uni}(0,1)^p$$
$$\beta_2 = \beta_2 / ||\beta_2||_1$$
$$Z'_i = \beta_2 Z_i$$
$$U_i = \mathbb{1}_{Z'_i < -3} + 2 * \mathbb{1}_{-3 \le Z'_i \le 1} + 8 * \mathbb{1}_{Z'_i > 1}$$
$$E_i \sim \mathcal{N}(0,1)$$

Finally, the label is defined as:

$$Y_i = 0.3 X_i \beta + 0.8 Z'_i + 0.2 + U_i E_i.$$

Turning to the corruption mechanism, for the dataset in which `Naive CP` achieves under-coverage, the corruption probability is defined as detailed in Appendix E.1. In the alternative dataset, in which `Naive CP` overcovers the response, the corruption probability is formulated as follows. For each sample, we subtract from $Z$ the minimal value between the 5% quantile and 0. Then, we divide by the 95% quantile of these values and multiply by 2.5. Next, we zero out all negative values. After that, we raise $e$ by the power of the negative of these values and take 1 minus this result. Lastly, we zero out all negative values and those that are greater than the 30% quantile. To obtain the corruption probabilities, we normalize these values between 0.2 and 0.9 and raise them to the power that leads to a 20% marginal corruption probability.

In Section 4.2 we employed a different corruption mechanism in which the weights are hard to estimate from $Z$. We begin by computing a complex function of $Z$:

$$T_i := \frac{\arctan\left(0.3\sqrt{6\sin^2(Z'_i)}\right)^{1/3} - 0.8\tanh\left(\cos\left(Z'^4_i\right)\right)}{0.5\sigma(Z'_i/2) + 0.5} + 0.5 + \sin\left(Z'^2_i/5\right) * \cos\left(Z'^4_i/8\right)$$

The rest of the process is similar to our default mechanism in Appendix E.1, except for zeroing all values for the samples with $T_i \le 1.2$ in addition to zeroing all negative values. Furthermore, instead of zeroing values that are lower than the 30% quantile, we zero the values that are lower than the 50% quantile. By incorporating $T_i$, which is a complicated function of $Z$, we turn the estimation process of $M \mid Z$ to be more challenging, which leads to inaccurate estimates of the weights.

# E EXPERIMENTAL SETUP

## E.1 GENERAL SETUP

Across all experiments, the data is divided into a training set (50%), a calibration set (20%), a validation set (10%) used for early stopping, and a test set (20%) for performance evaluation. For the `UI` method, we further split the original calibration set equally into a reference set (50%) and a calibration set (50%). Next, we normalize the feature vectors and response variables so that they have a zero mean and unit variance. In experiments with missing variables, we impute them using a linear model that is fitted on the variables that are always observed, out of $X, Y, Z$. This linear model is trained using samples from both the training and validation sets. For datasets that are not originally corrupted, the corruption probability is defined as follows. First, for the MEPS19 dataset, we fit a random forest model on the entire dataset to predict the 70% and 30% quantiles of $Y$ given $Z$, and we use their difference as the initial value. For the other datasets, we take $Z$ as the initial value; if $Z$ is multi-dimensional, we multiply it by a random vector to convert it into a scalar. We start by subtracting the minimum value between the 5% quantile and 0, then divide by the 95% quantile of these values, and multiply by 2.5. Negative values are then set to zero. Next, we raise $e$ to the power of the negative of these values and subtract the result from 1. Finally, we zero out any negative values and those below the 77% quantile. To obtain the corruption probabilities, these values are normalized to lie between 0.2 and 0.9 and then raised to a power that results in a 20%

marginal corruption probability. Thus, by definition, the average corruption probability is 20%. In every experiment, we train a base learning model and then wrap it with a calibration scheme. The learning model is designed to estimate the 5% and 95% conditional quantiles of $Y \mid X$. In Table 3, we summarize the models employed for each dataset across both tasks. For neural network models, we use an Adam optimizer (Kingma & Ba, 2015) with a learning rate of 1e-4 and a batch size of 128. The network architecture contains hidden layers with sizes 32, 64, 64, 32, with a dropout rate of 0.1, and uses leaky ReLU as the activation function. We used 100 estimators for both xgboost and random forest models. The networks are trained for 1000 epochs; however, training stops early if the validation loss does not improve for 200 epochs, at which point the model with the lowest validation loss is selected. We used the scikit-learn package (Pedregosa et al., 2011) to construct random forest models and the xgboost package (Pedregosa et al., 2011). The neural networks are implemented using the PyTorch package (Paszke et al., 2019). The hyperparameters we employed are the default ones unless stated otherwise. For `PCP`, we set the parameter $\beta$ to $\beta = 0.005$. In all experiments in which we employed `UI`, unless specified otherwise, we used a `Full+Linear` model for the label regression model $\hat{g}(x, z)$, in which a linear model is given both $Z$ and the output of a neural network model trained using $X, Z$. Moreover, the conditional errors were sampled using the linear clustering approach described in Section C.3.1 When applying a `Kmeans` clustering, we use the default number of clusters $k = 8$.

Table 3: The learning models used for each dataset.

| Dataset | Base learning model | Corruption probability estimator |
|---|---|---|
| **Facebook1 (facebook)** | Neural network | Neural network |
| **Facebook2 (facebook)** | Neural network | Neural network |
| **Bio (bio)** | Neural network | Neural network |
| **House (house)** | Neural network | Neural network |
| **Meps19 (meps_19)** | Random forest | Random forest |
| **NSLM (Yeager et al., 2019)** | XGBoost | XGBoost |
| **Synthetic datasets (Feldman & Romano, 2024)** | Neural network | Neural network |

### E.2 INACCURATE WEIGHTS STUDY

We generated 30000 samples from the synthetic datasets in Appendix D.3. As described in Appendix E, we split each dataset into training, validation, calibration, and test sets. We applied the error to the weights according to the setup and employed each method using the inaccurate weights. For the varying error setup, the errors were sampled independently of the data. The performance was computed for 30 random splits of the data in the constant error setup. For the varying error setup, we fix the training and calibration set and fix $X^{\text{test}}$ to $X^{\text{test}} = (2.6752, 1.2141, 2.0997, 4.4819, 3.9244, 4.1068, 4.9509, 1.9368, 4.8397, 1.6686)$ and $Z^{\text{test}}$ to $Z^{\text{test}} = (-2.9365, -3.4784, 1.3291)$, and generate 100K random response values $Y^{\text{test}}$ conditional on these values. The values we fixed for $X^{\text{test}}, Z^{\text{test}}$ were drawn from their marginal distribution. Since the calibration data, as well as $X^{\text{test}}, Z^{\text{test}}$ are fixed, the validity regions are deterministic and therefore can be computed accurately. The validity intervals of the varying errors setup are computed according to Theorem 3 as follows:

$$\delta_{\min} = -\frac{\delta_{\max}\tilde{\Delta}_{n+1} + W_{n+1}}{n + 1 - \tilde{\Delta}_{n+1}},$$

$$\delta_{\max} = \delta_{\min}\left(\frac{1}{\delta} + 1\right).$$

Above, $\delta$ is set to

$$\delta = \frac{\tilde{\Delta}_{n+1}W_{k^{\text{WCP}}} - \tilde{\Delta}_{k^{\text{WCP}}}W_{n+1}}{W_{n+1}k^{\text{WCP}} - (n+1)W_{k^{\text{WCP}}}}.$$

Theorem 3 states that `PCP` applied with any values of $\delta_{\min}, \delta_{\max}$ in this interval range is guaranteed to achieve a valid coverage rate.

### E.3 MACHINE'S SPEC

The resources used for the experiments are:

- **CPU**: Intel(R) Xeon(R) CPU E5-2683 v4 @ 2.10GHz, Intel(R) Xeon(R) Gold 5318Y CPU @ 2.10GHz, Intel(R) Xeon(R) Gold 6336Y CPU @ 2.40GHz.

- **GPU**: NVIDIA A40, NVIDIA TITAN X (Pascal), NVIDIA 2080 TI, NVIDIA RTX 2060 SUPER.

- **OS**: Ubuntu 20.04.6.

Experiments typically take at most 10 minutes to run, though actual times may vary with workload.

## F ADDITIONAL EXPERIMENTS

### F.1 WCP WITH INACCURATE WEIGHTS

We study the coverage rate attained by WCP when applied with various distributions of weight errors. We employ WCP on the two synthetic datasets described in Appendix D.3. In the first dataset, Naive CP achieves over-coverage, and in the second one, Naive CP undercovers the response. In Figure 6 and Figure 7 we display the validity regions of WCP with various distributions for the error of the weights. These figures show that the validity regions depend on the distribution of the error. Furthermore, it is indicated by the figures that the validity region is a small interval when Naive CP undercovers the response while when it overcovers, the validity region spans through the entire space except for one interval.

### F.2 PCP WITH INACCURATE WEIGHTS

In this section, we analyze the coverage rate of PCP when applied with various distributions of weight errors. We study the performance using the two synthetic datasets described in Appendix D.3. In the first dataset, Naive CP achieves over-coverage, and in the second one, Naive CP undercovers the response. Figure 9 and Figure 10 show the validity regions of PCP with various distributions for the error of the weights. These figures show the effect as observed in Appendix F.1.

### F.3 CAUSAL INFERENCE EXPERIMENT: NSLM DATASET

In this causal inference example, our goal is to estimate the uncertainty of individual treatment effects (Hernán & Robins, 2010). We utilize the semi-synthetic National Study of Learning Mindsets (NSLM) dataset (Yeager et al., 2019), which deals with behavioral interventions. Further details about the dataset can be found in Carvalho et al. (2019, Section 2), and Appendix D.2 outlines our adaptation for this dataset. Here, $X_i$ are the individual's characteristics, $Z_i$ are the privileged information, $M_i \in \{0, 1\}$ denotes the binary treatment indicator, and $Y_i(0), Y_i(1) \in \mathbb{R}$ denote the counterfactual outcomes under control and treatment conditions, respectively. In practice, we only observe one of them, $\tilde{Y}_i$, which equals to $Y_i(0)$ if $M_i = 0$ and to $Y_i(1)$ if $M_i = 1$. In this task, our goal is to estimate the uncertainty of the unknown response under no treatment $Y_{n+1} \equiv Y_{n+1}(0)$ at a pre-specified level $1 - \alpha = 90\%$. As explained in Feldman & Romano (2024), estimating uncertainty for $Y_i(0)$ is crucial since it can be used to construct a reliable prediction interval for the individual treatment effect (ITE), $Y_i(1) - Y_i(0)$, which is of great interest in many causal inference applications (Brand & Xie, 2010; Morgan, 2001; Xie et al., 2012; Florens et al., 2008). Furthermore, Feldman & Romano (2024) emphasizes that constructing valid prediction sets for $Y_{n+1}(0)$ is challenging due to the distribution shift between the observed control responses, which are drawn from $P_{Y(0)|M=0}$. In contrast, the test control response is drawn from $P_{Y(0)}$. Moreover, Feldman & Romano (2024) highlight the difficulty of constructing valid prediction sets for $Y_{n+1}(0)$, as it requires correcting the distribution shift between the observed control responses, which follow the distribution $P_{Y(0)|M=0}$, and the test control responses, which are drawn from $P_{Y(0)}$.

We display the performance of each calibration scheme in Figure 11. This figure indicates that Naive CP and Naive Imputation fail to achieve the nominal $1 - \alpha = 90\%$ coverage rate. In

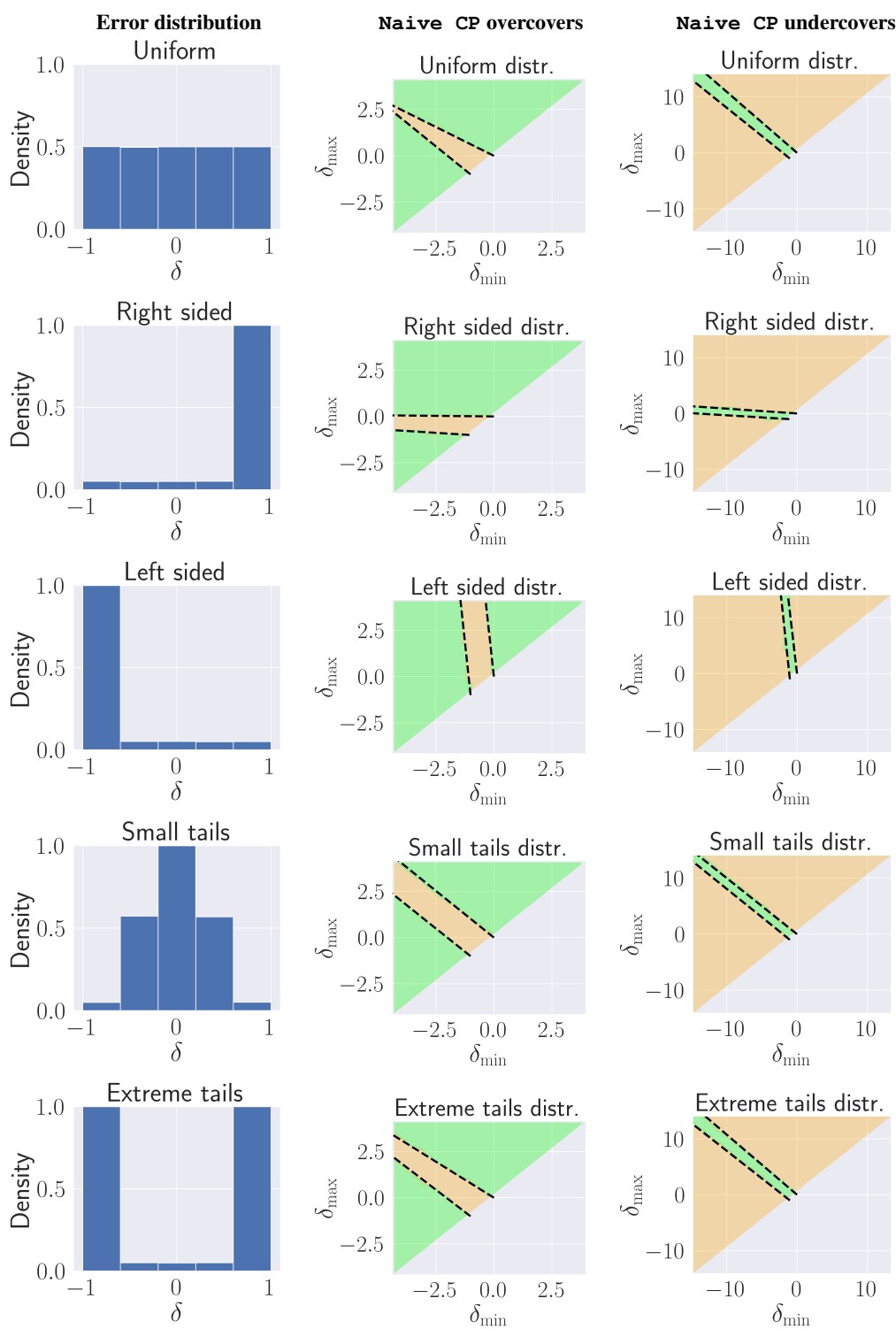

Figure 6: The validity regions of `WCP` applied with inaccurate weights along with the theoretical bounds from Theorem 3 displayed in dashed line. Here, the coverage rate is computed over random draws of 100K test responses $Y^{\text{test}}$ conditionally on the calibration set, $X^{\text{test}}$, and $Z^{\text{test}}$. Green: valid coverage, i.e., greater than the coverage rate of `WCP` with true weights; Orange: invalid coverage. Left: Distribution of the error. Mid: `Naive CP` achieves over-coverage. Right: `Naive CP` achieves under-coverage.

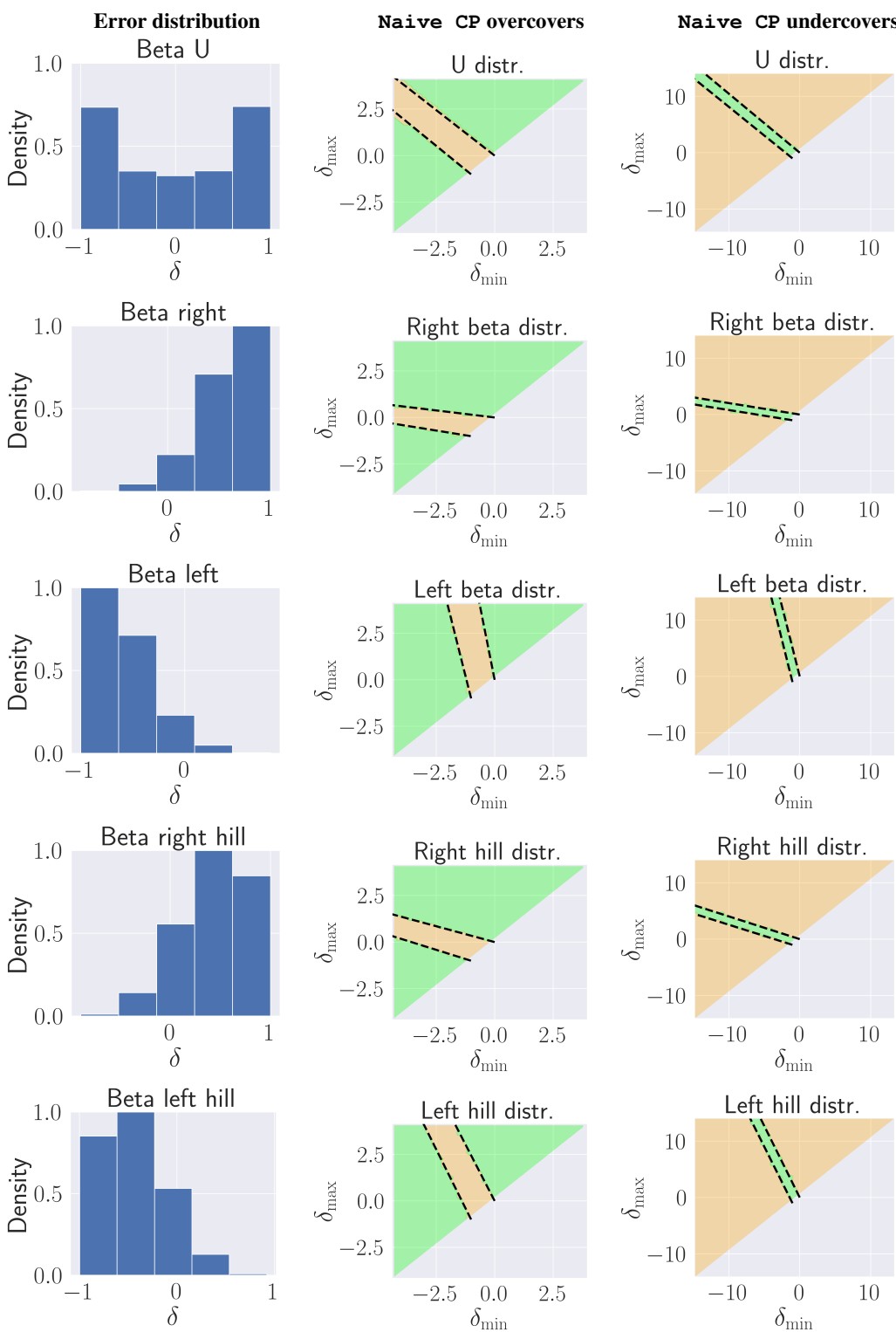

Figure 7: The validity regions of `WCP` applied with inaccurate weights along with the theoretical bounds from Theorem 3 displayed in dashed line. Here, the coverage rate is computed over random draws of 100K test responses $Y^{\text{test}}$ conditionally on the calibration set, $X^{\text{test}}$, and $Z^{\text{test}}$. Green: valid coverage, i.e., greater than the coverage rate of `WCP` with true weights; Orange: invalid coverage. Left: Distribution of the error. Mid: `Naive CP` achieves over-coverage. Right: `Naive CP` achieves under-coverage.

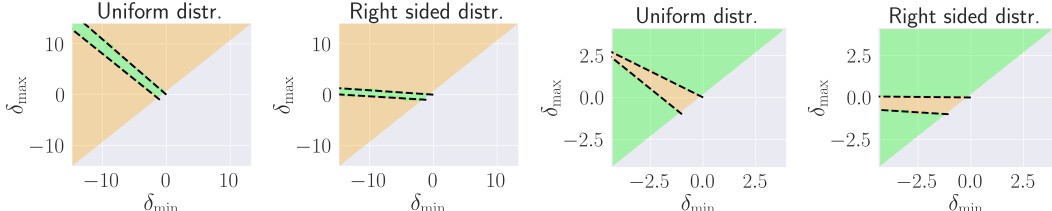

Figure 8: The validity regions of `PCP` applied with inaccurate weights along with the theoretical bounds from Theorem 3 displayed in dashed line. Here, the coverage rate is computed over random draws of $Y^{\text{test}}$ conditionally on the calibration set, $X^{\text{test}}$, and $Z^{\text{test}}$. Green: valid coverage region, i.e., greater than the coverage rate of `WCP` with true weights; Orange: invalid coverage region. Left: `Naive CP` under-covers the response. Right: `Naive CP` achieves over-coverage.

contrast, `PCP` attains a valid coverage rate, despite being employed with estimated weights. Moreover, our proposed `UI` constructs valid uncertainty sets as well, as guaranteed by our theory.

### F.4 IMPUTATION AND ERROR SAMPLING METHODS

We study the impact of the label regression model $\hat{g}$ of `UI` and the effect of the error sampling mechanisms from Appendix C.3.1 on the validity of the constructed prediction sets. For this purpose, we follow the setup detailed in Section 4 and use the same real datasets in a missing response setting. We apply `UI` with the following label regression models, aiming to achieve 90% coverage rate:

- `Linear`: where $\hat{g}$ is a linear function of both $X$ and $Z$, implemented using Scikit-learn's LinearRegressor (Pedregosa et al., 2011);
- `Full`: a neural network that takes $X$ and $Z$ as inputs;
- `Full+Linear`: a combined method in which a linear model is given both $Z$ and the output of the pre-trained `Full` model.

Figure 12 presents the performance of `UI` with the `Linear` model, showing that `UI` tends to overcover the response. This can be explained by the large estimation errors of the linear model, caused by its limited expressive power, leading to increased uncertainty in the imputed samples. This high uncertainty drives `UI` to construct large uncertainty sets.

In contrast, as shown in Figure 13, the `Full` model achieves a coverage rate that is closer to the nominal level when applied with linear or K-means clustering error sampling. However, the marginal error sampling approach, which does not condition on $Z$, leads to undercoverage. This highlights the importance of sampling errors conditionally on $Z$ to obtain valid prediction sets.

Finally, Figure 14 illustrates that the `Full+Linear` model tends to attain the target coverage when the linear or K-means clustering error sampling techniques are applied. Once again, the marginal error sampling strategy results in undercoverage, emphasizing that conditioning the error sampling on the privileged information is necessary to construct reliable uncertainty sets.

To conclude, these experiments demonstrate the importance of both an accurate label regression model $\hat{g}$ and a conditional error sampling mechanism to construct prediction sets that are valid in the missing response setup.

### F.5 EMPIRICALLY EVALUATING THE ASSUMPTIONS OF THEOREM 4

In this section, we assess whether the assumptions of Theorem 4 are satisfied in practice, by conducting two experiments following the protocol in Section 4.3.

In the first experiment, we evaluate the dependence between residual errors $Y^{\text{test}} - \hat{g}(X^{\text{test}}, Z^{\text{test}})$ and (i) the predictions $\hat{g}(X^{\text{test}}, Z^{\text{test}})$, (ii) the lower bounds $C^0(X^{\text{test}})$, and (iii) the upper bounds $C^1(X^{\text{test}})$ of the intervals produced by `UI`. We assess the dependence by computing the partial correlation (PC) conditional $Z^{\text{test}}$ between the terms. We emphasize that residual errors were used in place of $R^{\text{test}}$ because the latter is unavailable in practice. Likewise, since the true function $g^*$ is unknown,

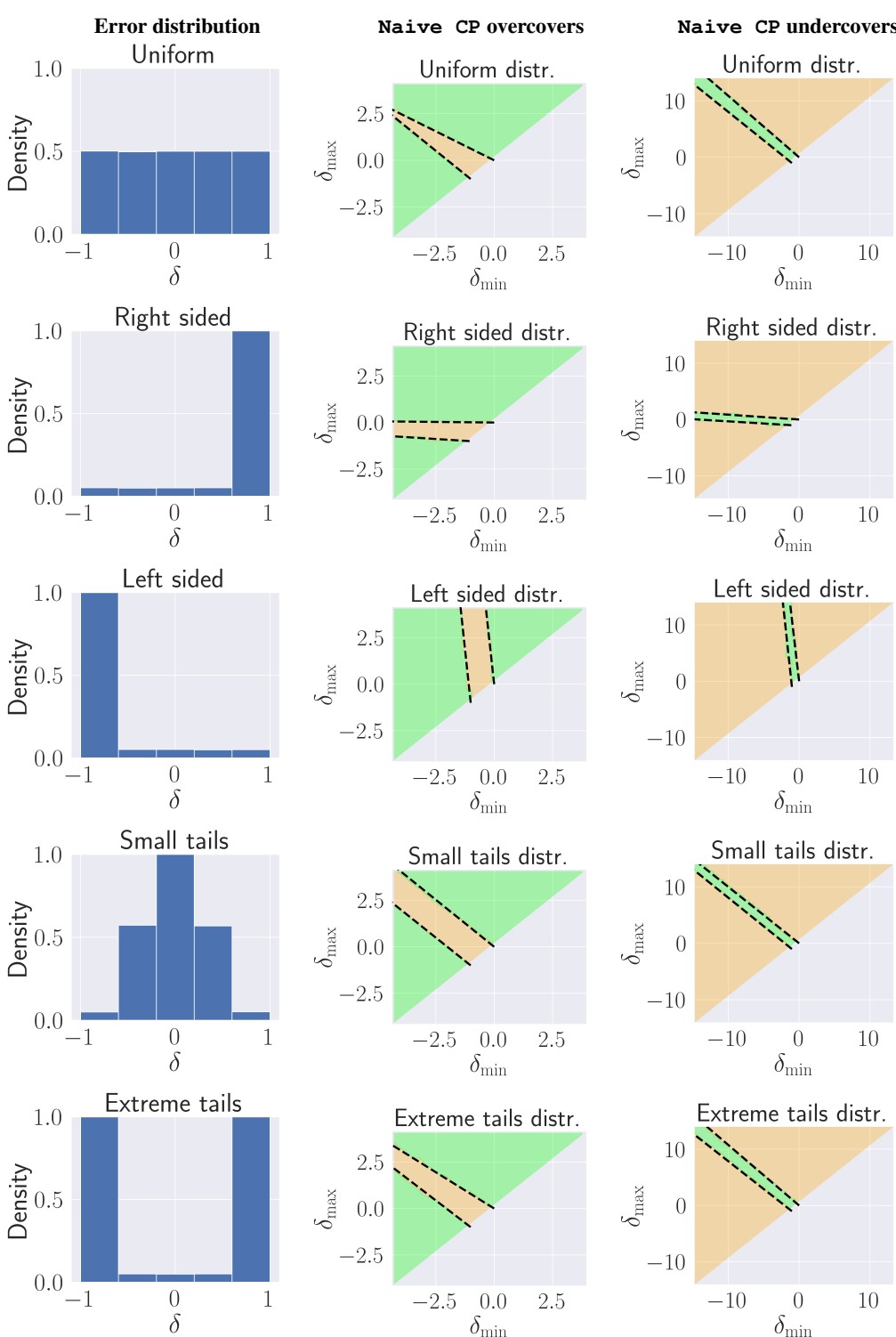

Figure 9: The validity regions of `PCP` applied with inaccurate weights along with the theoretical bounds from Theorem 3 displayed in dashed line. Here, the coverage rate is computed over random draws of 100K test responses $Y^{\text{test}}$ conditionally on the calibration set, $X^{\text{test}}$, and $Z^{\text{test}}$. Green: valid coverage, i.e., greater than the coverage rate of `PCP` with true weights; Orange: invalid coverage. Left: Distribution of the error. Mid: `Naive CP` achieves over-coverage. Right: `Naive CP` achieves under-coverage.

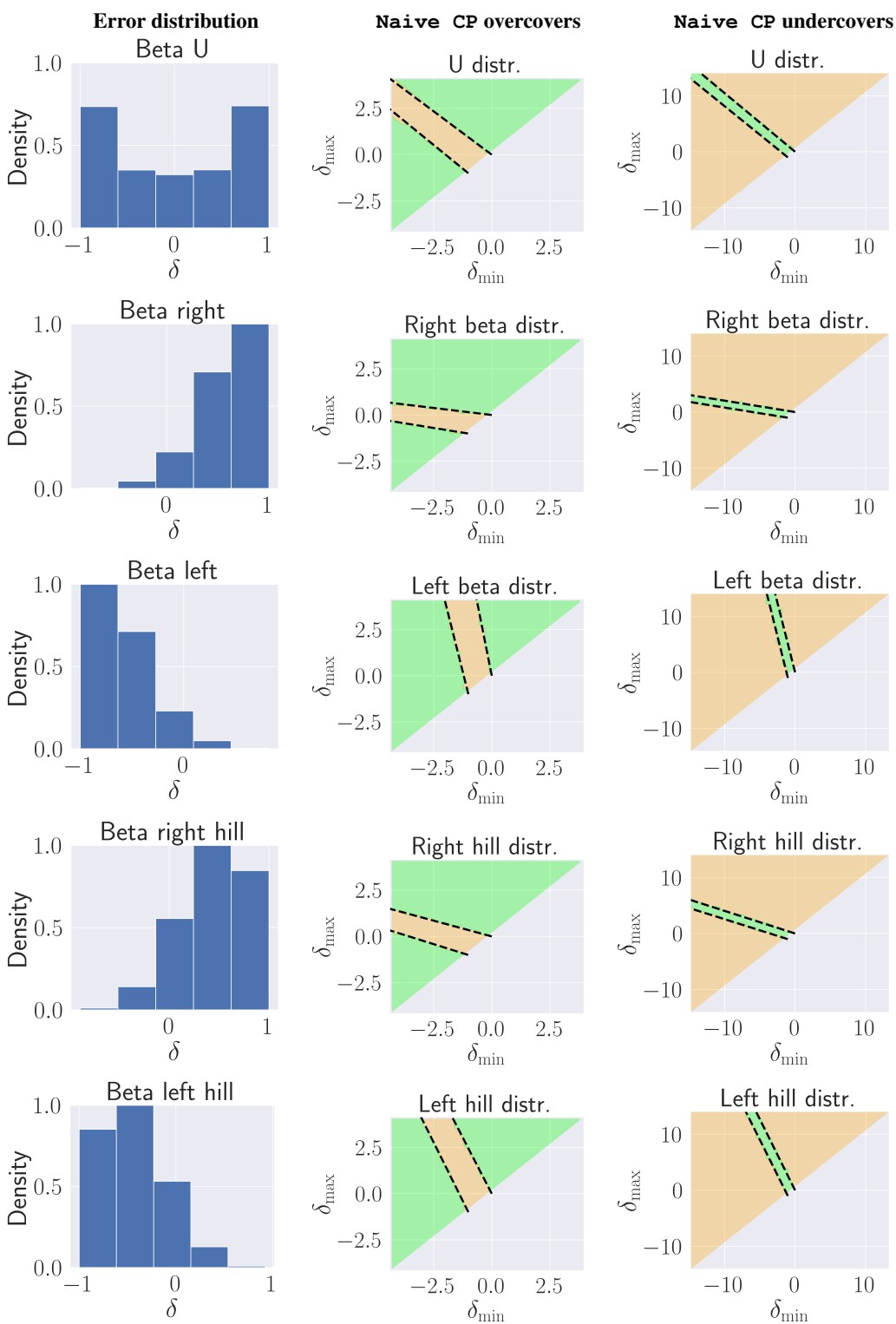

Figure 10: The validity regions of `PCP` applied with inaccurate weights along with the theoretical bounds from Theorem 3 displayed in dashed line. Here, the coverage rate is computed over random draws of 100K test responses $Y^{\text{test}}$ conditionally on the calibration set, $X^{\text{test}}$, and $Z^{\text{test}}$. Green: valid coverage, i.e., greater than the coverage rate of `PCP` with true weights; Orange: invalid coverage. Left: Distribution of the error. Mid: `Naive CP` achieves over-coverage. Right: `Naive CP` achieves under-coverage.

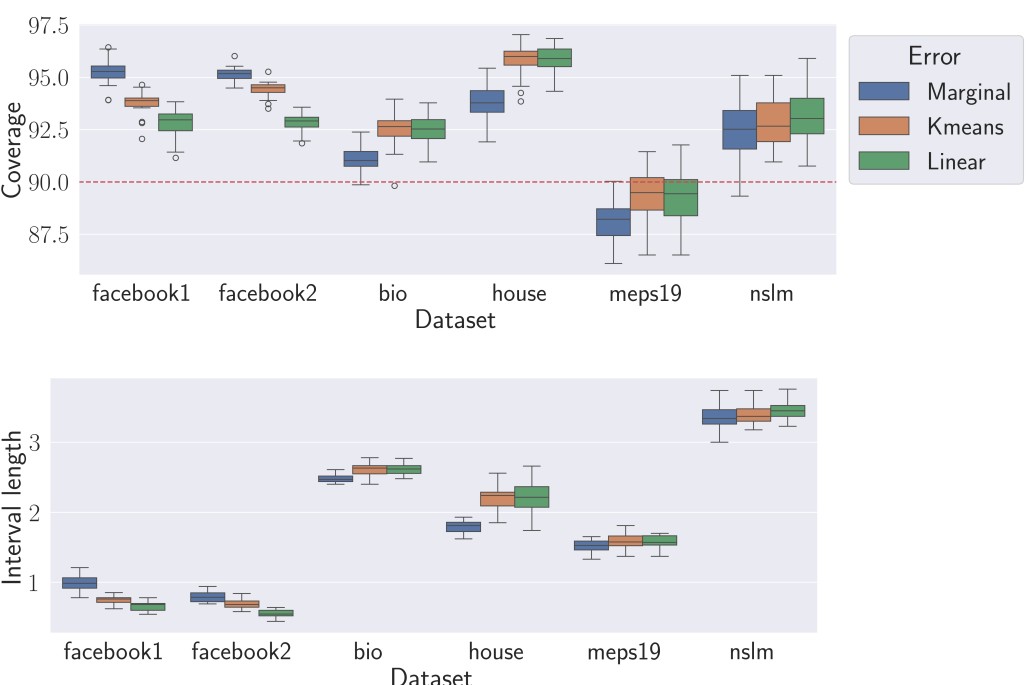

Figure 11: **NSLM dataset experiment.** The coverage rate and average interval length achieved by naive conformal prediction (`Naive CP`), conformal prediction with naive imputations `Naive Imputation`, `PCP` which estimates the corruption probability from $Z$, and the proposed method (`UI`). All methods are applied to attain a coverage rate at level $1 - \alpha = 90\%$. The metrics are evaluated over 30 random data splits.

Figure 12: **`Linear` regression model.** The coverage rate and average interval length obtained by `UI` with various error sampling methods. Performance metrics are evaluated over 30 random data splits.

we replaced it with its estimator $\hat{g}$. These replacements serve as fair alternatives for the unknown variables. For completeness, we also report the coverage rate attained by `UI` and the prediction error (MSE) of $\hat{g}$. All metrics were computed on the test set, and the results are averaged over 10 random data splits. The empirical results, summarized in Table 4, reveal that there is a non-negligible partial correlation between the interval endpoints and the residual errors. This suggests that the independence requirements of Theorem 4 are not exactly satisfied. Nevertheless, the intervals of `UI` still achieve the nominal coverage rate. This observation reveals the robustness of the `UI` procedure: even when the theoretical conditions are only approximately satisfied, it still constructs valid intervals.

Table 4: Metrics assessing the assumptions of Theorem 4.

| Dataset | Coverage | $\hat{g}$ **MSE** | $\hat{g}$ **PC** | $C^0$ **PC** | $C^1$ **PC** |
|---|---|---|---|---|---|
| Bio | $90.72 \pm 0.49$ | $0.56 \pm 0.00$ | $-0.02 \pm 0.00$ | $0.03 \pm 0.00$ | $-0.02 \pm 0.00$ |
| Facebook1 | $90.96 \pm 0.36$ | $1.84 \pm 0.21$ | $-0.07 \pm 0.05$ | $-0.02 \pm 0.04$ | $-0.03 \pm 0.05$ |
| House | $91.23 \pm 0.36$ | $0.71 \pm 0.03$ | $0.15 \pm 0.01$ | $0.18 \pm 0.01$ | $0.19 \pm 0.01$ |

In the second experiment, we analyze the setup where the PI is a weak predictor of $Y$. We follow the same experimental protocol in Section 4.3 and simulate this setup by adding random Gaussian noise

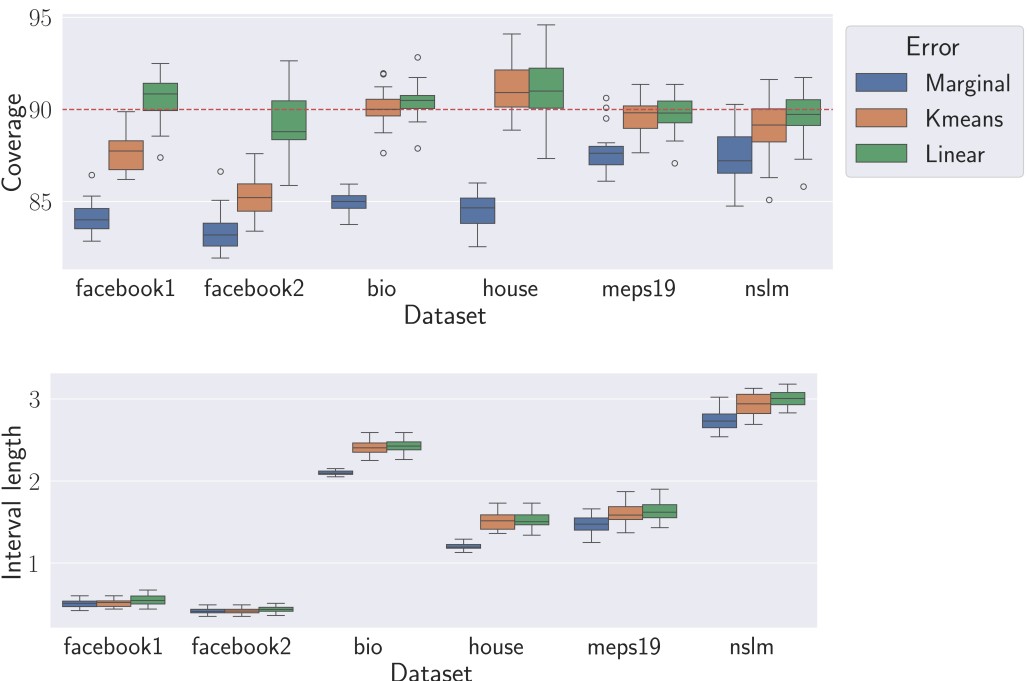

Figure 13: **Full regression model.** The coverage rate and average interval length obtained by `UI` with various error sampling methods. Performance metrics are evaluated over 30 random data splits.

to $Z$, with varying standard deviations. Figure 15 summarizes the performance of `UI` with these weak PIs, indicating that as the magnitude of the noise increases, the coverage rate of `UI` increases. That is, as the PI becomes a weak predictor of the labels, the uncertainty of the imputed labels increases, which, in turn, widens the resulting prediction sets. This experiment suggests that `UI` can still obtain a valid coverage rate even when the PI is a weak predictor of the labels.

### F.6    THE PERFORMANCE OF TRIPLYROBUST

We employ `TriplyRobust`, which combines `CP`, `PCP` that uses either the estimated corruption probabilities or the true ones, and `UI` on the datasets from Section 4.3. We present the coverage rate and interval length in Figure 16. This figure shows that `TriplyRobust` employed with `PCP` that uses estimated weights constructs wider intervals, which is in line with the results in Section 4.3, in which this version of `PCP` constructs wider intervals, as its approximations are not sufficiently accurate. Nevertheless, this figure reveals that combining the three approaches does not significantly harm the statistical efficiency of the predicted intervals, especially when using oracle or sufficiently accurate weights.

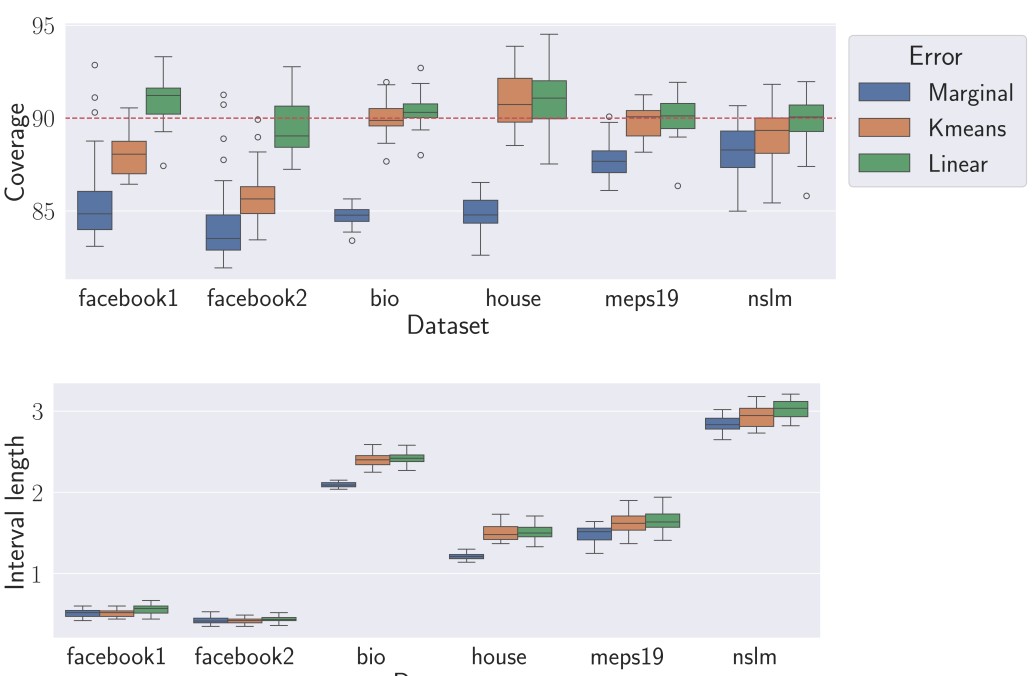

Figure 14: **Full+Linear regression model.** The coverage rate and average interval length obtained by `UI` with various error sampling methods. Performance metrics are evaluated over 30 random data splits.

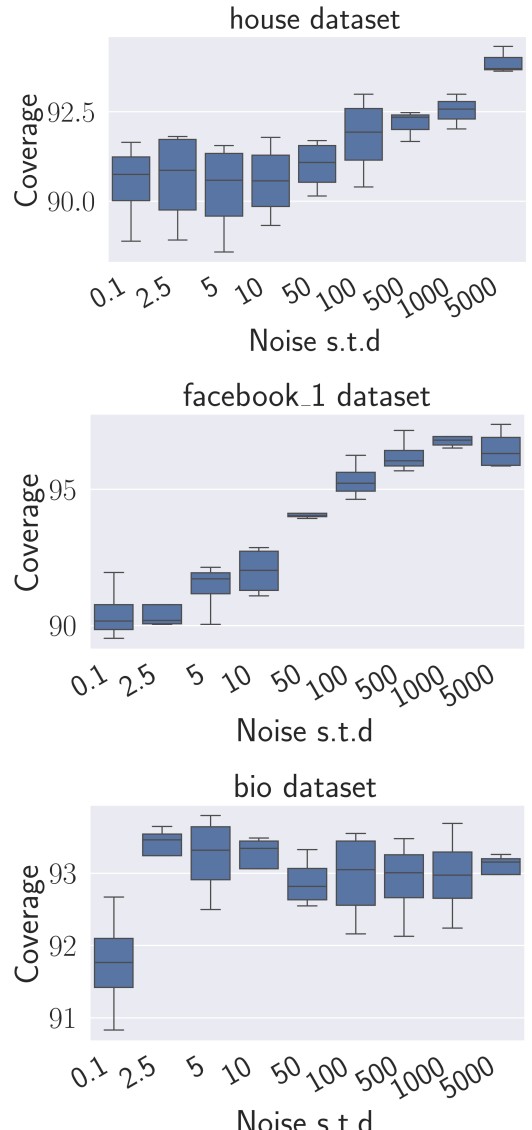

Figure 15: **Weak PI experiment**. The coverage rate as a function of the noise of the PI. The coverage rate is evaluated over 4 random data splits.

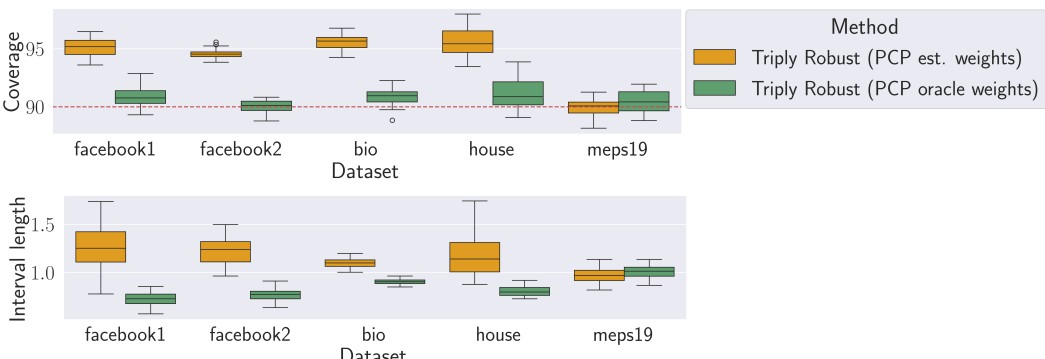

Figure 16: **Missing response experiment.** The coverage rate and average interval length obtained by `TriplyRobust` employed with `PCP` that uses either the estimated corruption probabilities or the true ones. Performance metrics are evaluated over 30 random data splits.

