# OpenReview forum: "Conformal Prediction with Corrupted Labels: Uncertain Imputation and Robust Re-weighting"
_ICLR.cc/2026/Conference — ICLR 2026 Poster_

### Official Review · Reviewer_9dC5 · 2025-10-27

**Soundness:** 3
**Presentation:** 3
**Contribution:** 3
**Rating:** 8
**Confidence:** 3

**Summary:**

This paper addresses the challenge of preserving the validity guarantees of Conformal Prediction (CP) when training and test data are non-exchangeable due to label corruption or missing information. The authors propose three complementary approaches to restore reliable coverage: Privileged Conformal Prediction (PCP), which reweights calibration samples using privileged information to correct for corrupted labels; Uncertain Imputation (UI), which imputes missing or noisy labels while injecting uncertainty to prevent overconfident intervals; and a Triply Robust CP method that ensures valid coverage if any of these assumptions hold. The paper provides theoretical proofs of coverage guarantees under each setting and supports them with synthetic and real-world experiments. Overall, it extends CP to corrupted-label scenarios and offers a practical framework for robust uncertainty quantification beyond the standard exchangeability assumption.

**Strengths:**

Conformal Prediction (CP) to data settings with label corruption and non-exchangeability. It also explores solutions for cases where the fundamental exchangeability assumption of traditional CP is violated. This is a problem that has received little attention in previous research.


The proposed combination of Privileged Conformal Prediction (PCP), Uncertain Imputation (UI), and Triply Robust CP creatively integrates ideas from conformal prediction, causal inference, and robust statistics, building a novel framework for uncertainty quantification under data corruption.


In terms of research quality, the theoretical analysis in the paper is rigorous, the assumptions are clearly stated, and precise coverage guarantees are provided under different robustness conditions. In terms of clarity, the paper is well-structured, with clear motivation and consistent notation. Although the technical content is deep, the core ideas are clearly expressed and easy to understand.

**Weaknesses:**

First, the distinction between exchangeability and i.i.d. should be made explicit early in the paper, particularly in the abstract. Currently, the text occasionally treats these terms as interchangeable, but this is conceptually inaccurate: exchangeability is a strictly weaker assumption than i.i.d., and making this distinction clear would improve both theoretical rigor and readability.

Second, the robustness assumptions underlying each proposed method (e.g., $((X,Y)\perp M|Z$) for PCP and accurate conditional modeling $(Y|Z)$ for UI) are relatively strong and may not hold in many real-world corruption processes. A more detailed discussion or empirical sensitivity analysis would help clarify how performance degrades when these assumptions are violated.

Third, while the theoretical analysis is rigorous, the implementation aspects (such as how to estimate the weighting function ($w(Z)$) or residual noise in high-dimensional settings) are under-explained and may limit reproducibility.

Finally, the experimental evaluation, though adequate, could be expanded to include additional diagnostics such as calibration error or coverage-interval efficiency, and more intuitive visualizations would help convey how weighting and imputation mechanisms restore validity.

**Questions:**

Exchangeability vs. i.i.d.: Please clarify the precise assumption used in your theoretical results. Some parts (e.g., the abstract) seem to equate exchangeability with i.i.d., though the former is weaker.

Assumption robustness:
How sensitive are PCP and UI to violations of their key assumptions? A short sensitivity or ablation analysis would clarify how performance degrades when these assumptions fail.

Implementation details:
Provide more detail on how the weighting function ($w(Z)$) and residual noise in UI are estimated in practice, particularly in high-dimensional settings, to improve reproducibility.

Evaluation metrics:
Consider adding calibration error or conditional coverage metrics to complement coverage and interval width, giving a fuller view of empirical performance.

Relationship to Literature: How do the authors' findings compare with the existing literature for CP with ambiguous ground truth in the calibration data, that is https://openreview.net/forum?id=L7sQ8CW2FY and https://proceedings.neurips.cc/paper_files/paper/2024/hash/d42a8bf2f40555d4a5120300f98c88f6-Abstract-Conference.html ?

---

> ### Author Response · Authors · 2025-11-18
>
> We thank the reviewer for their positive feedback and interest in our work. We thank the reviewer for considering our framework novel. In what follows, we address your comments in detail.
> > The exchangeability and i.i.d. assumptions
>
> We thank the reviewer for highlighting the importance of distinguishing between exchangeability and i.i.d. We agree that we should explicitly distinguish exchangeability and i.i.d. We referred to the i.i.d. setting in the abstract because it is more familiar to a broad audience. In the theorems, we assumed i.i.d. only when exchangeability alone does not suffice for the proofs, and otherwise rely on exchangeability. While we used these assumptions precisely and appropriately, we understand that using both terms can be confusing. We will revise the abstract and relevant sections to express this distinction and improve readability clearly.
>
>
> > The assumptions of PCP and UI
>
> Thank you for raising this important subject. The experiments in Appendix F.5 suggest that while the independence requirements of Theorem 4 are not exactly satisfied for the real datasets, UI still achieves the nominal coverage rate. This observation reveals the robustness of the UI procedure: even when the theoretical conditions are only approximately satisfied, it still constructs valid intervals. Regarding PCP, our experiments in Figures 2, 6-10, and 15 illustrate that PCP can attain a valid coverage rate as well under various noise distributions and levels.
> We will add this discussion to the updated manuscript. Thank you for helping us improve our paper!
>
> > Estimating the weighting function and the residual noise
>
> We thank the reviewer for bringing up this subject. All implementation details are provided in Appendix E, and our code is given in the supplementary material. The weighting functions used are listed in Table 3 of Appendix E.1. As the reviewer suggested, if the privileged information is high-dimensional, the sampling of residual error $E_i$ becomes more challenging. In Appendix C.3.1 we propose several approaches to alleviate this, such as clustering or distribution estimation. We will clarify these issues and our suggestions to handle them in the revised manuscript.
>
>
>
> > Additional experimental diagnostics
>
> We thank the reviewer for this valuable comment. We are uncertain about the definition of calibration error mentioned by the reviewer, as this work focuses on regression problems, and the standard metrics are coverage and interval length. We are certainly open to adding specific metrics if the reviewer finds they are critical for evaluating our proposals.
>
> In any case, as the reviewer suggested, we provided additional measurements that help understand the behaviour of our methods. In Table 4 of Appendix F.5, we measured the MSE of $\hat{g}$ and partial correlations evaluating the independence assumptions of Theorem 4. These metrics help validate that our theoretical conditions are approximately satisfied in practice. Additionally, our experiments in Figures 2, 6-10, and 15 examine the coverage of PCP under various conditions, including different error distributions, noise levels, and data-generating mechanisms. These experiments provide insight into how the methods perform across various settings.
>
> Thank you for suggesting the idea of intuitive visualization. We will include in the revised manuscript histograms of the imputed scores and the weighted scores to convey how they mimic the distribution of the test scores, explaining the coverage validity guarantee.
>
> > Relationship to literature
>
> We thank the reviewer for this excellent question and for pointing us to these interesting works. While both lines of work address corrupted calibration data, we see a limited immediate connection for two main reasons: first, these works focus on classification tasks, whereas our work addresses regression. Second, the problem formulations differ fundamentally: these papers assume multiple observed responses per sample are available during calibration, while we consider settings where responses are entirely unobserved (missing) or completely corrupted.
>
> That said, this connection sparks exciting research questions: could our uncertain imputation framework be extended to handle multiple noisy responses by treating them as a form of privileged information? Could their noise-learning techniques be utilized for weight estimation in our setting? Exploring whether our theoretical guarantees (Theorems 2-4) extend to the multiple-annotator setups, and whether their ambiguity-aware calibration methods could benefit from privileged information, would be valuable future work integrating these research directions.
>
> We appreciate the reviewer bringing this connection to our attention, and we will add a brief discussion of these works in our related work section to clarify the distinctions and potential future works.

---

### Official Review · Reviewer_wCwu · 2025-10-30

**Soundness:** 3
**Presentation:** 3
**Contribution:** 3
**Rating:** 6
**Confidence:** 3

**Summary:**

The paper addresses the label corruption in conformal prediction. While conformal prediction assumes exchangeability between the calibration points and the test point, if a subset of calibration points have corrupted labels (specifically if the corruption is non-random or dependent to a feature) then the exchangeability breaks. The paper is focused in the case that there is a privileged information either indicating the faulty label or carrying information about the true label.

For the case where the information indicates the corruption label, one existing approach - a.k.a. PCP - is to treat each calibration point as a test point and calibrate on the rest of non-corrupted points. With those scores we can find a threshold for the left out calibration point. Then the conformal threshold will be the 1 - beta quantile of those threshold. For this case the authors evaluate the robustness of PCP under various noise in the weights of the weighted conformal prediction. They further characterize under which cases does the PCP sets violate the coverage guarantee.

Additionally they also propose a label imputation setup by splitting the calibration and adding the label-estimation error to the labels.

**Strengths:**

1. The problem is applicable in many cases when there is any type of label noise.
2. The authors approached the problem in an organized way. They clearly break down the cases where the labels are faulty, and address each case separately.
3. The theoretical contribution of the paper is considerable.

In total while the problem is not clearly defined and solved I think the contribution in theory is above the standard for acceptance.

**Weaknesses:**

1. In general I could not connect the theorems in Section 3.1 to derive a robustness guarantee and a well delivered understanding of what the procedure is and what the guarantee would be. This is a shortcoming in the application as it is not clear what assumption holds.
2. Minor writing points: Line 176 the term indicator is used twice, maybe you can drop the second one.
3. Why the authors even discuss the setup with the constant noise on the weights? Isn’t it too unrealistic?
4. The definition of sufficiently accurate does not conform with the term used. Here the only requirement for the error is to be independent from conditional to Z.

**Questions:**

1. I can not understand the PCP setup. What is the Q(Z_i)? Do you compute WCP with 1 - alpha threshold for it? Does your theorem 1 hold for any values of beta?
2. Are you sure about line 190? Shouldn’t it be the training and calibration distribution?
3. I am not sure about it, but seems that the theorems in Section 3 are relying on the conditional assessment of error. If so, is this possible in real datasets? Because this means that you are intuitively assuming that you already have access to the clean label that you can estimate its corruption.
4. Is the entire requirement in Theorem 4 for the noise of the classifier to be independent from X conditional to Z? Does any classifier work? Is this even possible to be independent to X conditional to Z?

---

> ### Author Response · Authors · 2025-11-18
>
> We thank the reviewer for the valuable review. We appreciate the helpful suggestions and feedback. In what follows, we address your concerns in detail.
>
>
>
>
> > The theorems in Section 3.1.2
>
> Thank you for bringing up this crucial topic. Theorems 2 and 3 show that the robustness of PCP mostly depends on whether Naive CP undercovers or overcovers the response. The intuition is that as the weight estimation error ($\delta$) goes to $\pm\infty$, PCP produces intervals closer to those of Naive CP. In these theorems, we derive bounds for the weight error $\delta$, depending on the coverage rate of Naive CP, under which PCP attains valid coverage. Since, in practice, it is impossible to determine the coverage rate of Naive CP on clean, ground truth labels (we assume label corruptions), we cannot determine if PCP attains the desired coverage rate. Yet, our work provides a deeper understanding of how the estimation error of the weights affects the coverage rate. To the best of our knowledge, we are the first to show that the estimation error can be large, and a valid coverage can still be achieved in practice. We will clarify this motivation more explicitly in the revised manuscript.
>
> > The term indicator is used twice
>
> We thank the reviewer for highlighting this point. We will change the wording in the updated manuscript.
>
> > The setup of constant noise
>
> Thank you for raising this important subject. We introduced the setup of a constant error mainly for pedagogical purposes - to illustrate how an error in the weights affects the coverage rate of PCP. This simplified setting helps readers to understand the effect of the errors without getting distracted by technical subtleties. After establishing this intuition, we extend the analysis to the more general case where the weights are non-uniform.
>
> > The term “sufficiently accurate”
>
> We thank the reviewer for bringing up this comment. As the reviewer suggested, the term “sufficiently accurate” might be confusing, as there is no requirement for the magnitude of the error. The restriction on $\hat{g}$ is only the independence assumption. We will clarify this point in the main text.
>
> > The PCP setup
>
> We thank the reviewer for raising this point. $Q(Z_i)$ is computed by applying weighted conformal prediction (WCP) with target level $1- \alpha + \beta$ using the uncorrupted calibration samples. Since the weights are a function of $Z_i$, the quantile $Q(Z_i)$ is a function of $Z_i$ as well, and hence the notation. Theorem 1 (from the original PCP paper) holds for any value of $\beta$ in range $(0, \alpha)$. We will clarify these points in the text.
>
> > The distributions in line 190
>
> Thank you for highlighting this point. In our setup, we assume that the observed training/calibration distribution is different from the test distribution due to the corrupted labels. Therefore, we weigh the conformity scores using the likelihood ratio between the training/calibration and test scores, so that the weighted calibration scores ‘look exchangeable’ with the test score. In this paragraph, we used the ‘training’ distribution, implicitly assuming that the training and calibration sets follow the same distribution. We will update the text to “calibration and test distributions” and clarify this subject in the paper.
>
>
> > Access to true labels and the corruption indicator
>
> We are uncertain about the term ‘conditional assessment of error’ used by the reviewer. We would appreciate it if the reviewer could clarify its definition. In particular, the reviewer mentioned Section 3 (PCP), but there is no assumption on the clean labels requirement for estimating their corruption in this section. Perhaps the reviewer is referring to a different section? We would appreciate any clarification and would be happy to elaborate further on the assumptions of our framework.
>
> > The requirements of Theorem 4
>
> We thank the reviewer for raising this important point. Theorem 4 includes multiple assumptions. First, the estimation error should be independent of $g^*$ and $C$ conditional on $Z$. Second, the density of $Y | X,Z$ should peak inside the prediction set. Lastly, we assume that $Y$ is modeled as a deterministic function of $X,Z$ with an additive random noise. So, only classifiers that satisfy these requirements are guaranteed to obtain a valid coverage rate. Nevertheless, our experiments from Appendix F.5 suggest that the intervals of UI still achieve the nominal coverage rate even when these assumptions are not exactly satisfied. We will clarify this point in the revised manuscript.

---

> > ### Comment · Reviewer_wCwu · 2025-11-27
> >
> > **Setup of constant noise.** I now see the reason, and that is also very helpful, but I would suggest the authors to point out the exact reason prior to the example. It would be very helpful if it does not bring the same confusion that brought to me.
> >
> >  **Conditional assessment of error.** I think my confusion was that I thought the authors are assuming that there is an indicator that shows which labels are perturbed. I revisited the section and I think I was wrong. Sorry for the inconvenience in addressing this point.
> >
> > I am convinced with all the answers and I think the paper is interesting. Thanks for your reply again.

---

### Official Review · Reviewer_Ypqe · 2025-11-01

**Soundness:** 3
**Presentation:** 3
**Contribution:** 2
**Rating:** 6
**Confidence:** 3

**Summary:**

This paper analyzes the robustness of existing weighted and privileged conformal prediction methods under inaccurate weight estimation and introduces a new Uncertain Imputation approach for corrupted labels.

**Strengths:**

1- The paper presents a solid and technically sound theoretical analysis of conformal prediction methods under label corruption, with clear insights into how PCP and WCP behave when weight estimates are inaccurate.

2- The manuscript is clearly written, logically structured, and easy to follow despite the technical content.

**Weaknesses:**

Please check questions!

**Questions:**

1- While the paper provides solid theoretical analysis, it would strengthen the work if the authors could elaborate on the computational overhead of combining PCP, UI, and CP in the Triply Robust scheme

2-  The assumptions underlying Theorem 4,may not hold in many practical scenarios. It would be valuable for the authors to discuss how realistic these assumptions are and whether the proposed method remains approximately valid when they are violated.

3- It would be helpful if the authors clarified what predictive models were used in their experiments

4- Can the authors comment on whether their theoretical results extend to more general or data-dependent weight estimation errors, beyond the fixed bias or bounded error settings analyzed in Theorems 2 and 3?

---

> ### Author Response · Authors · 2025-11-18
>
> We appreciate your review of our work and thank you for the helpful suggestions and positive feedback. In what follows, we respond to your comments in detail.
>
>
>
> > The computational overhead of the TriplyRobust method
>
>
> We thank the reviewer for raising this point. Each calibration scheme, PCP, UI, and CP, produces a score threshold. TriplyRobust simply chooses the maximum among them. As a result, its computational overhead increases only by a constant factor compared to these methods.
>
>
>
>
> > The assumptions of Theorem 4
>
> We thank the reviewer for highlighting this crucial topic. To assess whether the assumptions of Theorem 4 are satisfied in practice, we conducted two experiments in Appendix F.5. These experiments suggest that the independence requirements of Theorem 4 are not exactly satisfied for the real datasets. Nevertheless, the intervals of UI still achieve the nominal coverage rate. This observation reveals the robustness of the UI procedure: even when the theoretical conditions are only approximately satisfied, it still constructs valid intervals.
>
>
>
>
> > The predictive models used in the experiments
>
>
> We thank the reviewer for this important comment. The predictive models used in our experiments are listed in Table 3 in Appendix E.1. For most datasets, we used a neural network. We employed Random Forest for the MEPS dataset and XGBoost for the NSLM dataset.
>
>
> > Extending the theoretical results in Theorems 2 and 3
>
> Thank you for this insightful comment. The results in Theorems 2 and 3 can indeed be extended to more general settings in which the weight estimation errors depend on the data. A conservative approach would be to quantify this dependence and derive a corresponding “worst-case” coverage bound that accounts for the dependence between error and data. We believe this idea is a promising future research direction.

---

### Official Review · Reviewer_7Q63 · 2025-11-01

**Soundness:** 3
**Presentation:** 2
**Contribution:** 2
**Rating:** 4
**Confidence:** 3

**Summary:**

In this paper, the authors investigate how to construct prediction sets with$1-\alpha$coverage for the test points, using datasets with corrupted labels. While the existing PCP method relies on a distribution shift$w(z)$to achieve the target coverage, its practical application is limited when $w(z)$ is not precisely known. To bridge this gap, the authors first analyze the theoretical behavior of PCP under scenarios where the shift$w(z)$deviates from the true value. Furthermore, they introduce a new approach called UI, which imputes corrupted labels and directly constructs valid prediction sets without requiring explicit knowledge of the distribution shift. Finally, the authors present extensive experimental results, demonstrating that the UI method achieves shorter prediction sets while maintaining the desired coverage guarantee.

**Strengths:**

First, the authors investigate the theoretical properties of the PCP method when the distribution shift$w(z)$deviates from the true value, thereby enriching the existing theoretical results. Second, the authors propose a novel UI method. By incorporating the idea of imputation, this method constructs reliable and effective prediction sets for test points without requiring the shift$w(z)$. The paper is clearly written and provides a new solution for constructing prediction sets using datasets with corrupted labels, thereby extending the existing body of work.

**Weaknesses:**

The paper could be strengthened by a more thorough discussion of the proposed UI method. For instance, Theorem 4 assumes that "the residual errors are independent of the predictions of $g^{*}$ and of$C^{UI}$given the PI $Z$." It would be important to clarify the practical scenarios in which this condition can be reasonably expected to hold. Furthermore, there is no clear evidence that the UI method consistently outperforms the PCP method. When facing a practical problem, what characteristics should one consider to determine which method is more appropriate? Finally, the experimental section would benefit from including the prediction set lengths of the TriplyRobust method for a complete performance assessment, and some notations could be revised for better clarity.

**Questions:**

1. Theorem 4 requires that "the residual errors are independent of the prediction of $g^{*}$and of $C^{UI}$ given the PI$Z$". However, under what specific settings can this condition be reasonably expected to hold?
2. When dealing with a practical problem, what specific characteristics should one consider to determine which method is more suitable? For instance, would the UI method outperform the PCP method when $Z$ is highly correlated with $Y$?
3. In line 318, is the residual $E_i$ computed solely for the uncorrupted samples, or for all samples in the reference set? If it's computed for all samples, the true $Y_i$ for the corrupted samples is unobservable. The notation here appears ambiguous.
4. In lines 343 and 344, regarding the condition for the peak of the conditional distribution within the interval $[a(x), b(x)]$, would the symbol "$\geq$" be more appropriately replaced by "$\leq$"?
5. In the experimental section, would the prediction interval lengths produced by the TriplyRobust method be significantly longer than those of the other methods?

---

> ### Author Response · Authors · 2025-11-18
>
> Thank you for your time and effort in reviewing our submission and providing valuable feedback and suggestions. In what follows, we address your concerns in detail.
>
>
>
> > Clarifying the assumptions in Theorem 4
>
> Thank you for raising this important subject. The first assumption in Theorem 4 requires the residual errors to be independent of the prediction of $g$ and of $C(X)$ given the PI $Z$. Intuitively, this means that the PI serves as a good proxy for $Y$. For example, in medical imaging, a pathologist's diagnostic report can act as privileged information that predicts tissue diagnosis. In this case, we require that the predictor that estimates $Y$ from the PI is accurate up to an error that is independent of $g^*, C$ conditional on $Z$. Observe that in our setup, we explicitly assume that the PI is a good predictor of $Y$, so the PI could truly assist in a good prediction of $Y$ - this is not a standard setup in prediction tasks. Importantly, our experiments from Appendix F.5 demonstrate that UI can obtain valid coverage even when the independence requirements of Theorem 4 are not exactly satisfied, revealing the robustness of our approach. We will clarify these points in the updated manuscript. Thank you for helping us improve our paper!
>
>
>
>
>
>
> > When UI is more appropriate than PCP
>
> We appreciate the reviewer’s insightful comment. PCP is reliable when the corruption indicator $M$ can be well estimated given $X,Z$. UI, on the other hand, is reliable when it is relatively easy to predict $Y$ given $X,Z$. UI does not compete with PCP, but rather complements it by relying on a complementary assumption. There are applications in which the PI better explains $Y$ than $M$. To the best of our knowledge, the proposed UI is the only method that provides a theoretical validity guarantee in this setup where the weights are unreliable. In cases where the user does not know which can be estimated accurately - $g(X,Z)$ or the weights- they can employ TriplyRobust to obtain a coverage guarantee that holds when at least one of them is sufficiently accurate. We will add this discussion to the manuscript.
>
>
>
>
>
>
> > The lengths of intervals produced by TriplyRobust
>
>
> We thank the reviewer for bringing up this comment. The prediction set lengths of the TriplyRobust method are presented in Figure 16 of Appendix F.6. This figure illustrates that TriplyRobust achieves the same interval length as PCP, as PCP generates the widest intervals in this setup. In the revised manuscript, we will refer to this figure in the main text.
>
>
> > The definition of the reference set using uncorrupted samples
>
>
> We thank the reviewer for highlighting this important issue. As the reviewer suggested, the reference set should be constructed using only the uncorrupted samples, that is: $\mathcal{E} =$  { $ E_i: i \in \mathcal{I}_3, Z_i =z, M_i = 0$ }. We will revise the manuscript accordingly.
>
> > The inequality in lines 343 and 344
>
>
> Thank you for raising this issue. You are correct that the inequality should be written as “$\leq$” and we will correct it in the revised manuscript. The proof itself remains valid, as the correct form of the inequality is already used at line 1236.

---

> > ### Comment · Reviewer_7Q63 · 2025-11-26
> >
> > Thank you for your detailed response and revisions. My concerns have been addressed, and  I have revised my score to reflect your updates.

---

### Meta-Review · Area_Chair_XJET · 2026-01-08

**Summary:**

The paper proposes a conformal set that preserves coverage guarantees when training and test data are non-exchangeable due to label corruption or missing information. They propose two different methods that rely on different kinds of privileged information to correct for the resulting non-exchangeability. They also propose a Triply Robust CP method that unifies the sets obtained by the two methods and vanilla CP, resulting in a set that ensures coverage if any of the underlying assumptions hold.

The reviewer found the paper interesting and well-written, and the problem important.

The reviewers also praised the rigorr of the theoretical results. The empirical evaluation sufficiently supports the claims of the paper, yet some reviewers had meaningful suggestions for improvement which I encourage the authors to consider.

The paper received overall positive ratings (4, 6, 6, 8). The questions raised by Reviewer 7Q63 who gave a score of 4 were sufficiently addressed in the authors' response, as well as all other reviewers' comments (see details below). Therefore, I recommend acceptance of the paper.

**Reviewer Concerns:**

The authors sufficiently addressed the comments raised by the reviewers. A common issue raised by multiple reviewers was regarding the assumptions of Theorem 4. The authors clarified these assumptions in their response. Their experiments in Appendix F.5 suggest that while the independence requirements of Theorem 4 are not exactly satisfied for the real datasets, their still achieves the nominal coverage rate in practice. The authors clarified also the question about exchangeability and i.i.d. assumptions, and provided additional details about estimating the weighting function and the residual noise. I suggest the authors to make the assumptions even more explicit in the main text in the final version and integrate the discussion from the rebuttal.

**Reviewer Scores:**

I believe that all reviewers who gave a positive score (6,6,8) would maintain their high score. Reviewer 7Q63 would have likely increased their score.

---

### Decision · Program_Chairs · 2026-01-26

Accept (Poster)